# Cingulate microstimulation induces negative decision-making via reduced top-down influence on primate fronto-cingulo-striatal network

Satoko Amemori[1,2,4], Ann M. Graybiel [3] & Ken-ichi Amemori [1,4] ✉

The dorsolateral prefrontal cortex (dlPFC) is crucial for regulation of emotion that is known to aid prevention of depression. The broader fronto-cingulo-striatal (FCS) network, including cognitive dlPFC and limbic cingulo-striatal regions, has been associated with a negative evaluation bias often seen in depression. The mechanism by which dlPFC regulates the limbic system remains largely unclear. Here we have successfully induced a negative bias in decision-making in female primates performing a conflict decision-making task, by directly microstimulating the subgenual cingulate cortex while simultaneously recording FCS local field potentials (LFPs). The artificially induced negative bias in decision-making was associated with a significant decrease in functional connectivity from cognitive to limbic FCS regions, represented by a reduction in Granger causality in beta-range LFPs from the dlPFC to the other regions. The loss of top-down directional influence from cognitive to limbic regions, we suggest, could underlie negative biases in decision-making as observed in depressive states.

Major depressive disorder (MDD) is a prevalent psychiatric disorder, affecting approximately 280 million people and 5% of the adult population worldwide[1]. MDD patients exhibit maladaptive emotional regulation and difficulties in effectively implementing adaptive strategies, suggesting that problems in the regulation of emotion are at the core of these cardinal symptoms of MDD[2]. The dorsolateral prefrontal cortex (dlPFC), which has long been implicated in cognitive functions, such as switching attention[3,4], working memory[5,6], and categorical learning[7], is also thought to be a center of emotion regulation[8,9]. The interareal interaction between the cognitive dlPFC and the limbic system is a possible critical factor for the protective mechanism against the development of MDD[2,10]. Especially, the fronto-cingulo-striatal (FCS) network, which includes the fronto-striatal circuit as the cognitive system[11] and the cingulo-striatal circuit as the limbic system[12,13], has been implicated in so-called negative processing bias[14,15]

in MDD patients[16,17], who tend to react negatively to emotionally evocative stimuli[18]. Fronto-cingulate[19] and fronto-striatal[20,21] interactions have been implicated in both cognition and emotion, highlighting the potential role of dlPFC in emotional regulation. In clinical studies, when healthy individuals confront affective challenges, an augmentation in fronto-cingulate coupling has been reported[22,23]. In contrast, individuals with MDD exhibited diminished fronto-cingulate connectivity[24], alongside abnormal fronto-striatal connectivity[20,21]. Specifically, the interaction among FCS network has been explored through transcranial magnetic stimulation (TMS) on the dlPFC, commonly used for MDD treatment[10]. The antidepressant effectiveness of the dlPFC activation is linked to their anticorrelated activities of the subgenual anterior cingulate cortex (sgACC)[25]. However, the neuronal mechanism of how the cognitive dlPFC regulates the cingulo-striatal system has remained largely unclear.

[1]Institute for the Advanced Study of Human Biology (ASHBi), Kyoto University, Kyoto, Japan. [2]Japan Society for the Promotion of Science, Tokyo, Japan. [3]McGovern Institute for Brain Research, Department of Brain and Cognitive Sciences, Massachusetts Institute of Technology, Cambridge, MA, USA. [4]These authors contributed equally: Satoko Amemori, Ken-ichi Amemori. ✉e-mail: amemori.kenichi.7s@kyoto-u.ac.jp

Potential candidates for such interareal interactions in the cortico-cortical and cortico-striatal networks have been extensively studied in non-human primates, including the suggestion that cognitive functions could be subserved by neuronal synchronization[26,27] in the fronto-striatal[7] and fronto-cingulate[3] networks. Such interareal synchronization has been assessed by coherence analysis of the local field potentials (LFPs) in these regions[28–30], and entrainment has been assessed by use of Granger causality (GC)[31,32]. For the interaction between cognitive and visual systems, an important set of studies on functional connectivity of fronto-parietal[4,5] and visual cortical areas[33,34] has demonstrated that directional influences measured by the GC can be represented by beta oscillatory activity[35,36]. The function of beta oscillations has been extensively explored, uncovering their involvement in motor control[37,38], attention[38], and decision-making[39]. Recently, the direction of the beta signaling was specifically found to be a correlate of the 'top-down' (i.e., feedback) direction of the cortical hierarchy as defined by its anatomy[32,40–42]. In the primate striatum, beta-band oscillatory activity has been found to represent decision-related variables[36], and the pattern of the beta oscillations at particular sites was found to parallel stimulation-induced negative bias elicited from those sites[35]. These findings prompted us to test the possibility that beta synchronization could also subserve top-down processing in the FCS network and that reduction of top-down processing of the network could result in impaired emotion regulation of the cognitive regions.

To address this possibility, we performed a functional intervention on a critical node of the cingulo-striatal regions to induce, experimentally, a negative bias in decision-making, and we examined the changes in beta oscillation signaling from the cognitive dlPFC to the limbic regions that occurred as a result. The sgACC was selected as the target for microstimulation due to its critical role in emotional modulation in MDD. In MDD patients, the sgACC exhibited elevated metabolic activity, which decreased with successful antidepressant treatment[43]. Deep brain stimulation of the sgACC effectively relieved symptoms in treatment-resistant depression[44,45]. Consistent with the clinical observations, primate neurophysiology studies revealed that sgACC stimulation in marmosets induced a negative bias in approach-avoidance tasks[46], accompanied by an increase in skin conductance, suggesting effects on cardiovascular activity[47]. Based on these findings, we conducted microstimulation on the sgACC, while macaque monkeys performed the approach-avoidance conflict task[48], a task that has been used to quantify how negative bias processing affects decision-making in both humans[49,50] and non-human primates[35,45,48,51].

While we performed microstimulation, we simultaneously recorded LFPs[52] from the multiple sites in the FCS network. Our targets within the FCS network included the dlPFC as the cognitive area and the pregenual anterior cingulate (pACC) and sgACC as cingulate areas known to have reciprocal connectivity[53]. Additionally, we targeted the dorsal part of the striatum[35], which is recognized for receiving projections from the dlPFC[54] and pACC[51]. Similarly to the effect of microstimulation of the pACC[48] and striatum[35], here we found that microstimulation of the sgACC successfully induced a negative bias in decision-making. Notably, this effective microstimulation significantly attenuated the directional beta-band influence from the dlPFC to both the pACC and striatum. These findings suggest that the activity of the cognitive dlPFC has a crucial function in regulating negative emotional bias processed by the limbic system. Furthermore, these findings suggest a striking parallel between the prefrontal influence on limbic networks and 'top-down' regulation of frontoparietal visual cortical networks[32,40]. The loss of directional top-down influence from the dlPFC to the cingulo-striatal regions could elicit dysfunction in emotion regulation, one of the core symptoms of MDD.

## Results

### Microstimulation of the sgACC induced negative bias in conflict decision-making

Two monkeys (S and P) were trained to perform an approach-avoidance (Ap-Av) decision-making task[48] (Fig. 1a). Specifically, in this task, a visual cue consisting of abutted red and yellow horizontal bars appeared after a 2-second pre-cue period. The lengths of the bars signaled the offered amounts of reward (red bar) and the offered pressure of air-puff delivered to the monkey's face (yellow bar). After the 1.5-second cue presentation period (i.e., decision period), targets, a square for avoidance and a cross for approach, appeared above or below the compound cues, their positions changing randomly for each trial. Within 3 s (i.e., response period), the monkey was required to report its choices by controlling a joystick to move the cursor to one of the targets. If the monkey chose the cross target (Approach or Ap choice), an airpuff with the strength indicated by the yellow bar was delivered, followed by a liquid reward of the amount indicated by the red bar. If the monkey chose the square target (Avoidance or Av choice), both the airpuff and reward were omitted. In the trained monkeys, the monkeys' decisions were systematically determined by the offered sizes of reward and airpuff (Fig. 1b), as reported before[48,50]. We calculated the reaction time (RT) as the time between target onset and target acquisition. The RTs were longer for Avoidance (Av) choices than for the Approach (Ap) choices, with the peak RTs occurring in the conflict conditions (Fig. 1c). Standard deviations in decision-making were also high in the conflict conditions (Fig. 1d).

To perform simultaneous microstimulation and recording of spikes and LFPs from multiple FCS regions, we implanted 43 electrodes in the right hemisphere of monkey S (3 in the sgACC, 12 in the dlPFC, 16 in the pACC, and 12 in the striatum) and 32 electrodes in the left hemisphere of monkey P (5 in the dlPFC, 6 in the pACC, 6 in the sgACC, and 15 in the striatum) (see Methods for details) (Fig. 1e, f). The location of each electrode was identified by postmortem histological reconstruction of the electrode tracks (Supplementary Fig. 1). We performed 38 stimulation experiments and 74 recording-only experiments. Recording-only experiments were conducted before each microstimulation experiment without moving the electrodes, allowing us to record LFPs from all stimulation and recording sites (Fig. 1f). For the microstimulation experiments, we selected one of the sgACC electrodes (Fig. 1e) and used it for microstimulation (frequency at 200 Hz, 70–100 µA), and the others were used for the simultaneous recordings. Stimulation sessions consisted of three blocks: *Stim-off* (200–250 trials before microstimulation), *Stim-on* (200–250 trials with microstimulation), and *Follow-up* (100-250 trials after microstimulation). We performed microstimulation for 1 s from the start of the decision period at every trial in the *Stim-on* block (Fig. 1a).

To quantify the effects of microstimulation, we measured the difference between decision matrices of the *Stim-off* block and the *Stim-on* block and derived the sizes of increases in the Ap choices (i.e., %ΔAp) and the Av choices (i.e., %ΔAv) (Fig. 1g). We set a 5% increase as the threshold for distinguishing 'effective' from 'non-effective' sessions for each sgACC stimulation site[20] (see Methods and Supplementary Fig. 2). Among the 38 stimulation experiments, microstimulation increased the Av by more than 5% in ten sessions (red circles in Fig. 1e, 10/38, 26.3%), defined as negative effective sessions. The microstimulation also increased the Ap by more than 5% in three sessions (blue circles in Fig. 1e, 3/38, 7.8%), defined as positive effective sessions. The proportion of negative effective sites was significantly greater than that for the positive effective sites (Chi-square test, $P = 0.033$; Fisher exact test, $P = 0.032$), suggesting that the functional intervention of the part of sgACC activities could induce a negative bias in decision-making.

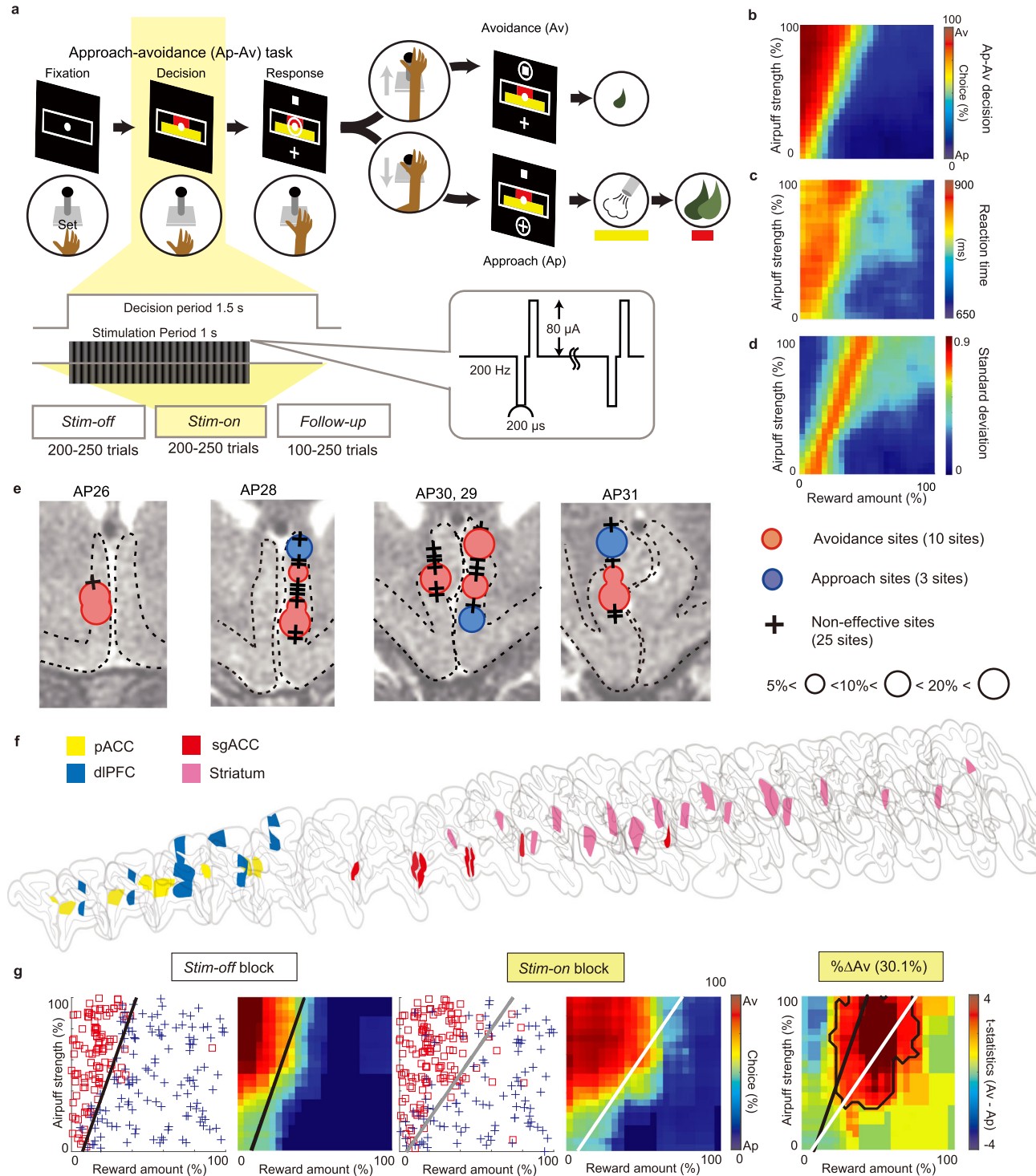

**Fig. 1 | Microstimulation of sgACC induced negative bias in decision-making.**
**a** The approach-avoidance (Ap-Av) task. **b** The grand average of Ap-Av choices was mapped on the decision matrix and spatially smoothed. The x-axis indicated the offered reward size, and the y-axis indicated the offered airpuff size. **c** The grand average of reaction times between the target onset and acquisition was shown. **d** The grand average of the standard deviation of Ap-Av choices was shown. **e** The distribution of the stimulated sites in the sgACC was mapped on coronal MRI images. Red and blue circles indicate the sites at which microstimulation induced changes in Av and Ap choices, respectively. The size of each circle indicates the size of the effects. A black cross indicates the site at which no effect was found. **f** We simultaneously recorded LFPs from the pACC (yellow), sgACC (red), dlPFC (blue) and striatum (pink). **g** An example of a negative effect in sgACC microstimulation was shown. Ap and Av choices during *Stim-off* and *Stim-on* blocks were represented by a decision matrix on the left and middle panels, respectively. A spatially smoothed decision matrix was also shown on the right of each decision matrix. The difference between the choice pattern in the *Stim-off* and *Stim-on* blocks was shown on the right panel. The zone showing significant difference was surrounded by a black line (*P* < 0.05, two-sided Fisher's exact test). The decision boundaries of the *Stim-off* and *Stim-on* blocks were indicated by black and gray/white lines, respectively.

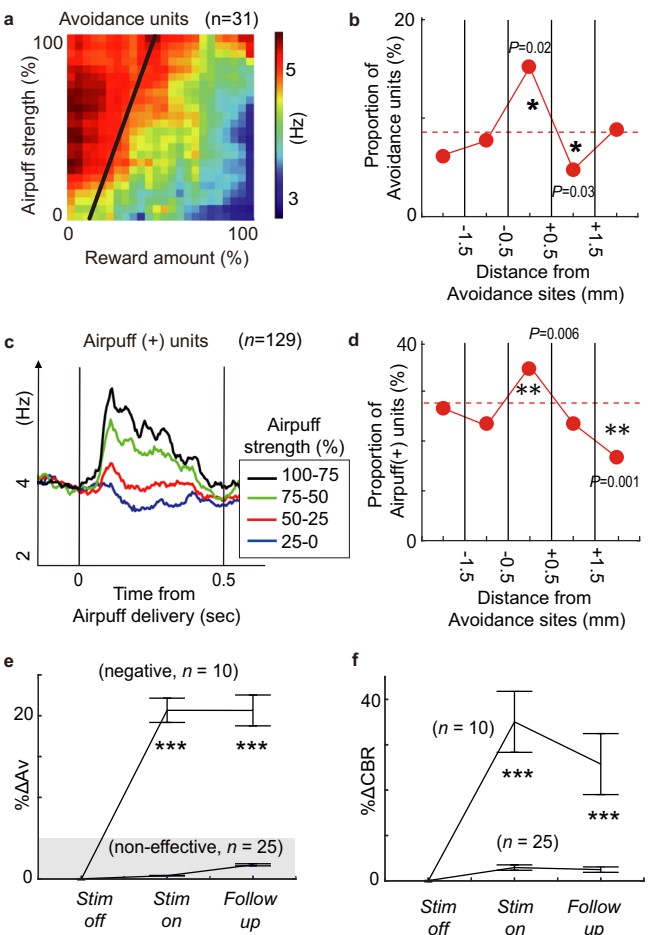

idea that the sgACC may be a key node in regulating saliency and arousal[55], essential for recognizing and responding to important stimuli[56]. The pupillary responses induced by the microstimulation were significantly smaller than those induced by airpuff delivery (Supplementary Fig. 3f, g). Because the effect of the microstimulation on the physiological response was significantly smaller than that induced by the airpuff, we concluded that the reflexive aversive reaction to microstimulation was too weak to influence Ap-Av decision-making.

We recorded spike activities from the implanted electrodes around the stimulation sites in the recording-only sessions and then examined the features of neurons recorded in this zone (Methods). Units with cue-period activity positively correlated with upcoming avoidance choices were classified as 'avoidance neurons' ($n = 31$, Fig. 2a). Within the 1-mm bins around the negative effective sites, the frequency of observing the 'avoidance neurons' was significantly greater than in other bins (Fisher exact test, $P = 0.02$) (Fig. 2b). Similarly, units with outcome-period activity positively correlated with the strength of airpuff were classified as 'airpuff (+) neurons' ($n = 129$, Fig. 2c), and the frequency of observing the 'airpuff (+) neurons' around the negative effective sites was significantly greater than in other bins (Fisher exact test, $P = 0.006$) (Fig. 2d). The spatial correlations between negative effective sites and the neuronal response patterns were not observed for the other type of units (Supplementary Fig. 4). These results suggest that activation of 'avoidance neurons' and 'airpuff (+) neurons' in the sgACC could serve as part of the network that is causally involved in negative bias in conflict decision-making.

We found that the negative bias in decision-making persisted after effective microstimulation. To measure this effect of the microstimulation, we calculated the increases in Av choices (%ΔAv) between *Stim-off* and *Follow-up* blocks for effective (Supplementary Fig. 5a) and non-effective (Supplementary Fig. 5b) sessions. We compared the magnitude of %ΔAv between effective ($n = 10$) and non-effective ($n = 25$) sessions (Fig. 2e). The mean %ΔAv for the *Stim-on* and *Follow-up* blocks was significantly larger than the 5% threshold in negative effective sessions ($n = 10$), indicating that negative biases in decision-making were sustained in the *Follow-up* block. By contrast, the mean %ΔAv for non-effective sessions ($n = 25$) remained consistently below the 5% threshold. When the stimulation was ineffective, the monkeys did not change their decision-making in either the *Stim-on* or *Follow-up* blocks (Fig. 2e). We used a conditional logit model[29] to implement logistic regression as a way to estimate the negative bias in decision-making. This model infers the subjective evaluation process by performing a logistic regression to derive the relative sensitivities towards reward and punishment (Methods). To characterize such relative sensitivities, we used a cost-benefit ratio (CBR), defined as the ratio of the weights for reward and punishment in the conditional logit model. We compared the CBRs in the *Stim-off*, *Stim-on* and *Follow-up* blocks to quantify the degree of negative bias in decision-making (Fig. 2f). The result found was that CBR was significantly higher in both *Stim-on* and *Follow-up* blocks compared to the *Stim-off* block.

**Fig. 2 | Spatial correspondence between neuronal features and stimulation effects. a** Population activity of avoidance neurons during the decision period mapped to the decision matrix ($n = 31$). The black line indicates the decision boundary derived by logistic regression. **b** Proportion of avoidance neurons was shown for 1-mm bins around avoidance sites. The results were expressed as a percentage, representing the number of avoidance neurons divided by the total number of recorded neurons in each bin. The asterisk indicates the bin in which the proportion of each neuron was significantly different from that of aggregated over all tracks (two-sided Fisher's exact test, *$P < 0.05$). **c** The time course of population activity of airpuff (+) neurons ($n = 129$) was represented according to the strength of the airpuff. The zero time point indicates the time of airpuff delivery.
**d** Proportion of airpuff units was shown for 1-mm bins around the negative effective sites, as illustrated in **b** (two-sided Fisher's exact test, **$P < 0.01$). **e** Mean percentage increase in avoidance choices (%ΔAv) for each block. The means of %ΔAv for the negative stimulation sessions ($n = 10$) were significantly larger than those for the non-effective sessions ($n = 25$) during the *Stim-on* and *Follow-up* blocks (two-sided two-sampled t-test, ***$P < 0.001$). Data are presented as mean values ± SEM.
**f** Mean percent increase in cost-benefit ratio (%ΔCBR) for each block. The means of %ΔCBR for the negative stimulation sessions were significantly larger than those for the non-effective sessions during the *Stim-on* and *Follow-up* blocks (two-sided two-sampled t-test, ***$P = < 0.001$). Data are presented as mean values ±SEM.

To determine the extent to which the microstimulation directly drives the aversive state, we examined the autonomic responses induced by effective microstimulation. Thus, we conducted an additional experiment in which we applied microstimulation to three negative effective sites while the monkeys performed a simple fixation task (Supplementary Fig. 3a). We observed a significant increase in pupil size without inducing any eye movements during the microstimulation (Supplementary Fig. 3b-e). Consistent with previous studies[47], these findings suggest that the activity of the sgACC plays a causal role in regulating physiological responses. This supports the

## Beta-band modulation of the FCS network coincides with stimulation-induced negative bias

To search for neural signals associated with negative bias, we recorded LFPs from the FCS network while the monkey performed the Ap-Av decision-making task (Fig. 1f). To identify task-related LFPs, we calculated peak frequencies of the spectral power density of all baseline-subtracted LFPs in the four regions and found that many peaks were in the beta range (13−30 Hz) (Supplementary Fig. 6a). The grand average of the power spectrum of all baseline-subtracted LFPs also showed beta oscillations in the four regions (Supplementary Fig. 6b). Based on these findings, we focused on beta oscillations in the FCS network.

We analyzed a total of 3716 LFPs recorded during the recording sessions or the *Stim-off* block of the microstimulation experiments.

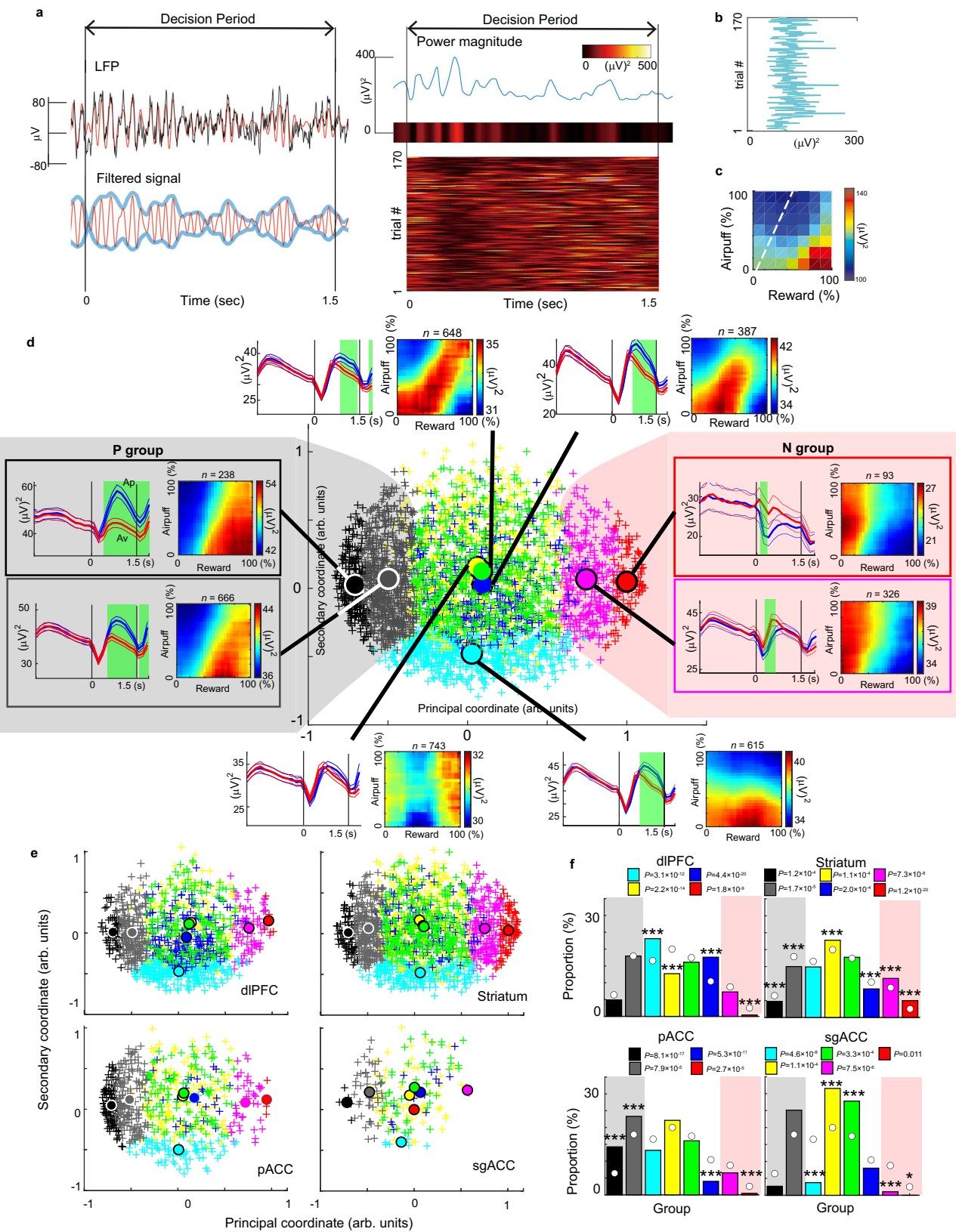

Specifically, 188 were recorded from the sgACC, 1696 from the striatum, 680 from the pACC, and 1152 from the dlPFC, all of which were utilized in the subsequent analyses. In previous work, we demonstrated that a subset of striatal beta-band LFPs were associated with Ap-Av choices[35]. We thus examined the relationship between the Ap-Av decision and the magnitudes of LFPs recorded in this experiment. To

extract the beta magnitude for each trial, we used a band-pass filter and computed the time course of beta power using the difference between the upper and lower envelopes (Fig. 3a). We then averaged the beta power over the decision period (Fig. 3b) and projected this onto a decision matrix, which we called 'the beta response' (Fig. 3c). The beta response of each channel shows how the beta magnitude

**Fig. 3 | Information encoded in beta responses in the FCS network. a** The black line in the left top panel indicates the LFP activity of a dlPFC site during the decision period, and the red line indicates its band-pass-filtered activity. The blue lines in the left bottom panel indicates the upper and lower envelopes of the filtered LFP. The right top panel shows the power magnitude for the trial, and the right bottom panel shows the trial-by-trial power magnitudes for each trial in a session, with the x-axis indicating the time and the y-axis indicating the trial number. **b** The average power magnitude for each trial. **c** The beta response matrix. **d** The results of MDS were presented at the center. Each cross represents an individual LFP with color indicating its assigned group. Eight circles are located at the center of each group. On the left, the population activities of the P group (consisting of LFPs in the gray and black groups) were shown by shaded gray. Similarly, the population activities of the N group (consisting of the pink and red groups, shaded pink area) were shown on the right. In each panel, the temporal response of population activities was shown on the left. Each line indicates the population mean ± standard error of the mean (SEM) of the power magnitude for the upcoming Av choice (red) and the Ap choice (blue). The green area indicates when the means for the Ap and Av choices were significantly different (two-sided t-test, $P < 0.05$, Bonferroni corrected over time bins). The population activity of the beta responses was shown on the right. **e**, MDS and clustering for the beta responses in each region. Each cross and circle were illustrated similarly to the center panel of **d**. **f** The proportion of each group in each region. An asterisk indicates a significant difference between the proportion of the group in each region and the proportion of the group across all regions, which is represented by the white circle (two-sided Fisher's exact test, ***$P < 0.001$, *$P < 0.05$).

varied for each offered reward and airpuff and represents the feature of each LFP in relation to the Ap-Av decision-making. To characterize the decision-related features of all beta responses, we employed multi-dimensional scaling (MDS) on the population of all beta responses (Supplementary Fig. 6c, d), and we performed an unbiased clustering analysis in the feature space (see Methods).

With the clustering, we could identify eight groups of beta responses (Fig. 3d; Supplementary Fig. 6e) that exhibited distinct features in the decision matrix. To probe better the relationship between the identified groups and behavior, we conducted MDS with a series of behavioral parameters (Supplementary Fig. 6e), including positive and negative expected utilities (Eutils). The MDS analysis showed that positive and negative Eutils were located at opposite ends of the MDS space, suggesting that the value of the principal component axis (principal component value, PCV), which represents the primary characteristic of the beta responses, is closely related to the expected utilities of the decision. Among the eight groups, the black and gray groups had the lowest and the second lowest PCVs (Fig. 3d) and were categorized as P (positive) group. The pink and red groups had the highest and the second highest PCVs and were categorized as the N (negative) group. The clustering further demonstrated distinct groups of LFPs, indicating specific activations for the low airpuff offer (cyan group in Fig. 3d), along with other groups showing activations for the decision boundary (blue and green groups in Fig. 3d) and the group showing activity for low and high reward offers (yellow group in Fig. 3d). Acknowledging the continuity rather than the distinct separation between similar clusters, we quantified predictive accuracy using posterior probabilities. Remarkably, 79.7% of the P group and 76.1% of the N group had posterior probabilities exceeding 75% (Supplementary Fig. 7a), suggesting that the clustering process was efficient for a substantial portion of the data classified into N and P groups. Notably, in the third-dimensional axis, the green, blue and yellow clusters are distinctly segregated (Supplementary Fig. 7b).

The distribution of beta response groups varied across regions in the FCS network (Fig. 3e), suggesting that beta oscillations may serve distinct functions in different brain regions. While the dlPFC showed an average proportion of the P group compared to other regions, it also exhibited a high percentage of the cyan and blue groups (Fig. 3f), which are associated with negative offered airpuff and conflict decision-making (Fig. 3d). This result suggests that the dlPFC may not be solely responsible for Ap-Av decision-making, but may, instead, participate in the regulation of conflict decision-making and motivation[49]. The striatal beta-band LFPs had a significantly higher proportion of the N group (Fig. 3e, f) than did other regions[35,50], whereas the sgACC had a significantly lower proportion of the N group (Fig. 3e, f). The P group had a substantial presence in all regions in the FCS network (Fig. 3e, f), and this group was more evenly distributed across the FCS regions than the N group.

We further employed a representational similarity analysis (RSA)[57,58] to explore the similarity of information processing among the four structures we recorded (Supplementary Fig. 8a, b). The analysis showed that cortical areas (sgACC, dlPFC, and pACC) exhibited distinct activation patterns for high utility, whereas the striatum showed varied activation for middle and low utilities, indicating contrasting regional representations between the striatum and cortices (Supplementary Fig. 8c, d). The RSA thus evaluated the striatum as different from the three cortical areas, similar to the clustering procedure results that showed the striatum contained a substantial number of the N group (Fig. 3e, f). Thus, the RSA repeatedly showed the regional specificities in the distribution of different types of beta responses in the FCS network.

Additionally, we performed regression analyses to determine the representation of the beta responses (Supplementary Fig. 9). For each channel, we further performed the all-possible subset regression with selected explanatory variables (offered reward size, offered airpuff size, expected utility, reaction time, and frequency of omission error) (see Methods) to investigate the information encoded in each LFP. The regression analysis demonstrated that 495 LFPs encoded positive offered reward or positive expected utility (Supplementary Fig. 9). Importantly, all of these 495 LFPs (495/495 = 100%) were categorized as belonging to the P group by the unbiased clustering analysis (Supplementary Fig. 9a, b). Similarly, we found that 208 LFPs encoded negative offered reward or negative expected utility, and the majority of them (202/208 = 97.1%) were categorized as being in the N group (Supplementary Fig. 9a, b). These results showed that the P groups exhibited activation for positive expectation of reward and utility, while the N group encoded negative expectation of reward and utility. Taken together, these analyses suggest that the P group might have a relatively prominent role in communicating the Ap-Av decision variable across different brain regions of the FCS network.

To investigate the impact of microstimulation on beta signals within the FCS network, we focused on the change in beta responses during sessions in which the sgACC microstimulation was effective in influencing decision-making in monkeys. We simultaneously recorded 57 task-related LFPs from the FCS network (i.e., dlPFC, sgACC, pACC and striatum) during ten effective sessions. To analyze the recorded LFPs, we employed MDS on the combined data from both recording-only and stimulation sessions. Subsequently, we applied clustering analysis to assign each LFP to one of the eight groups (Fig. 4). We first examined how effective microstimulation changed the population average of the PCVs of the LFPs. Interestingly, we observed a significant increase in mean PCV in the *Stim-on* and *Follow-up* blocks compared to levels in the *Stim-off* block (upper panel in Fig. 4a). By contrast, no significant change in mean PCV was observed in the 25 non-effective sessions during the *Stim-on* and *Follow-up* blocks (lower panel in Fig. 4a). These results, obtained in causal experiments, demonstrate that microstimulation of the sgACC, particularly when effective in producing behavioral effects, modulated the overall representation of beta responses within the FCS network.

Importantly, the stimulation effect on beta-band activity in the FCS network was primarily observed in the P group of LFPs, which represents a positive offered reward or expected utility. The effective microstimulation significantly reduced the proportion of beta

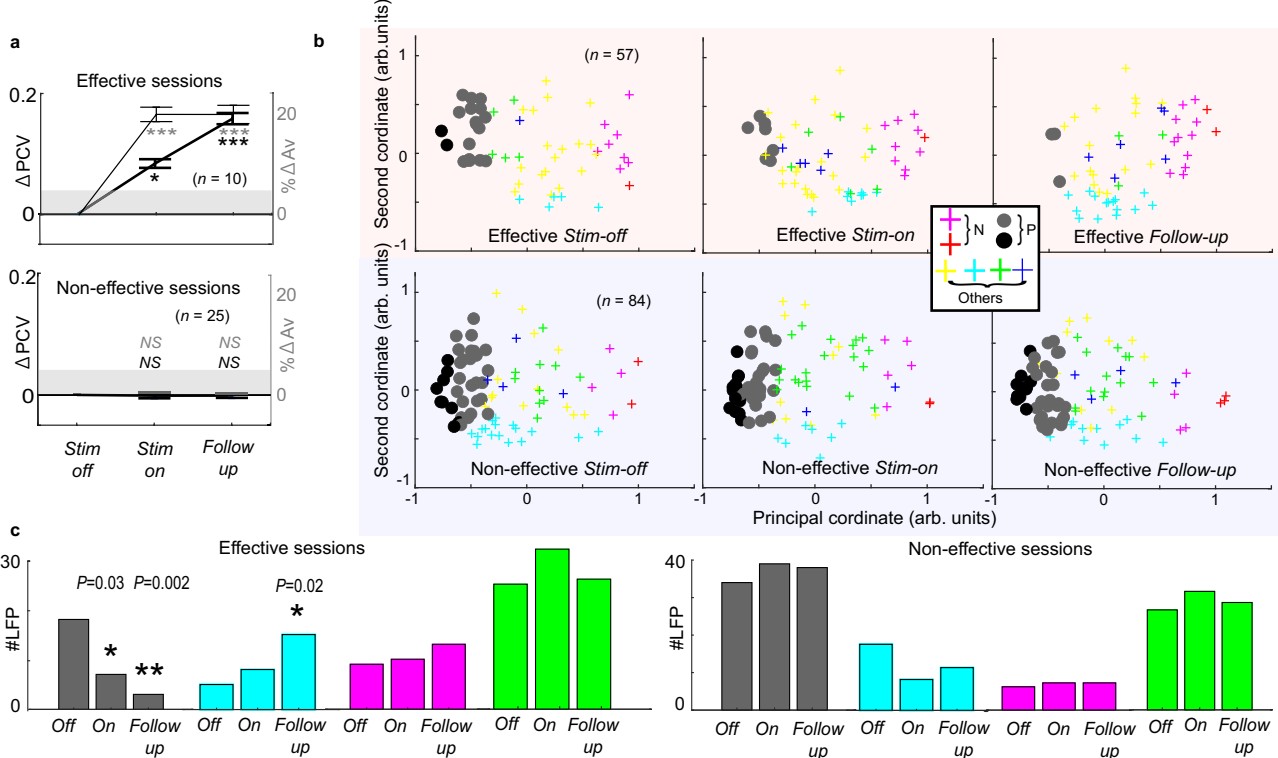

**Fig. 4 | Effective microstimulation diminished the beta representation of the P group.** **a** Mean changes in principal coordinate value (ΔPCV) (thick black) from *Stim-off* to *Stim-on* and *Follow-up* blocks were plotted along with the percent increase in Avoidance choices (ΔAv) (thin gray). The top panel shows results for the negative effective sessions, and the bottom panel shows those for non-effective sessions. The means of ΔPCV and ΔAv increased significantly during the *Stim-on* and *Follow-up* blocks for the negative effective sessions (*n* = 10) (two-sided t-test, ***P* < 0.001), while those for the non-effective sessions (*n* = 25) did not (*P* > 0.05). Data are presented as mean values ±SEM. **b** MDS plots of LFPs in *Stim-off* (left), *Stim-on* (middle), and *Follow-up* (right) blocks were shown for the negative effective

sessions (top panels) and non-effective sessions (bottom panels). Thick black and gray circles indicate the LFPs in the P group. Crosses indicate those classified in the other groups, with colors corresponding to each group. **c** The numbers of LFPs classified as each group were shown for *Stim-off* (*Off*), *Stim-on* (*On*), and *Follow-up* blocks. The left and right panels show results for the negative effective and non-effective sessions, respectively. The proportions of the gray group were significantly reduced during the *Stim-on* and *Follow-up* blocks for the negative effective sessions, and the cyan group increased the proportion in the effective *Follow-up* block (two-sided Fisher's exact test, **P* < 0.05, ****P* < 0.001). For all groups, no significant change in the non-effective sessions (*P* > 0.05) was observed.

responses classified as the P group (Fisher's exact test, *P* < 0.05), whereas non-effective microstimulation did not (Fig. 4b, c). The cyan group increased the proportion in the effective *Follow-up* block (*P* < 0.05). No significant change was observed in other groups (*P* > 0.05). These results suggest that effective microstimulation of the sgACC could induce a negative bias in decision-making by changing the representation of the beta oscillation in the FCS network, particularly for the P group.

We next examined how the sgACC microstimulation affected the beta responses that had been categorized as the P group in the *Stim-off* block. We found that effective microstimulation influenced their position in the MDS space. Compared to the positions in the *Stim-off* block, the positions in the *Stim-on* and *Follow-up* blocks shifted to the right (Supplementary Fig. 10a), whereas the non-effective microstimulation did not induce this effect (Supplementary Fig. 10b). We also derived an Ap-Av tuning index by subtracting the spectrum for the Av choices from that for the Ap choices. Compared to the Ap-Av tuning index in the *Stim-off* block, the effective microstimulations significantly decreased them in the *Stim-on* and *Follow-up* blocks, rendering the animals more avoidant (Supplementary Fig. 10a). By contrast, the non-effective microstimulation did not have a significant effect (Supplementary Fig. 10b). Our analyses thus demonstrated that the effective sgACC microstimulation, which induced a negative bias in decision-making, particularly influenced the representation of the P group. As the P group was recorded in all regions in the FCS network,

we reasoned that the sgACC microstimulation could have influenced the interareal interaction of the FCS network and tested for this possibility.

## Beta Granger causality was attenuated by stimulation-induced negative bias

Convincing evidence supports the view that local rhythmic synchronization can lead to interareal synchronization between the frontal cortex and other regions of the brain[5,30,32], including the striatum[7,28]. Interareal synchronization within beta-frequency bands in LFP recordings has been thought to mediate the top-down control of visual attention[4,26,32], and to have anatomical correspondence with the laminar patterns of origin and termination of the connections[41,42]. Based on this evidence, we hypothesized that LFP synchronization in low-frequency bands (i.e., alpha and beta) might lead to interareal synchronization between the dlPFC and cingulo-striatal components of the FCS network. We further hypothesized that dysfunction of this network could result in diminished interareal interaction. To test these predictions, we examined the Granger Causal Influences (GCIs) of pairs of LFPs recorded from the four regions in the FCS network studied. Interareal synchronization between different brain regions can be quantified by the degree of coherence between pairs of sites (Methods). First, we examined the frequency band in which the interareal coherence was observed. Our analysis of coherence in all pairs (*n* = 3172) showed peaks around 7 Hz and 20–25 Hz, but no peaks in the

gamma range (Supplementary Fig. 11a). We thus focused on the coherence in the alpha and beta ranges (5–30 Hz). Synchronous oscillations in the low-frequency band were consistently present during the *Stim-off* block in both monkeys (P and S) (Supplementary Fig. 11b).

We, therefore, calculated frequency-specific GCIs between pairs in the different regions of the FCS network to determine whether directional influences at these frequencies could be found (Methods; Supplementary Fig. 12). GCIs were averaged over both alpha (5–13 Hz) and beta (13–30 Hz) frequencies (Fig. 5). To characterize the directional influences, we calculated the directional asymmetry index (DAI) for each pair in the FCS network and for each frequency in the alpha and beta ranges (Methods). For each connectivity, we produced an arrow in the circular graph and showed them in Fig. 6 for alpha and beta frequencies, separately. Remarkably, the directional influences in the *Stim-off* block showed that the signaling in the FCS network was unidirectional in both the alpha and beta ranges (Fig. 6a), indicating that the FCS network could have a hierarchical position, resulting in the dlPFC being at the top, the pACC at the second level, the sgACC at the third level, and the striatum at the lowest level.

We next tested the prediction that sgACC microstimulation could attenuate top-down influences from the dlPFC. To calculate directional influences in the network, we compared the difference of GCIs across pairs of LFPs (Fig. 6b, c). The microstimulation significantly reduced the top-down influence in the beta oscillation while enhancing the bottom-up influence mediated by alpha oscillation in the FCS network. Specifically, in the alpha range, GCIs from the sgACC and striatum indicated a pronounced strengthening of the bottom-up influences. In the beta range, the top-down influences originating from the dlPFC were significantly attenuated (Fig. 6b).

We further explored the causal relationship between the network-level changes and the behavioral alterations. Notably, the behavioral changes induced by the stimulation exhibited temporal accumulation (Supplementary Fig. 13a), a feature consistently reported in our previous studies[35,48]. Importantly, the changes in the FCS network began earlier than the increase in avoidance choices (Supplementary Fig. 13b). Granger causality analyses demonstrated that the network changes Granger-cause the increase in avoidance choices (Fig. 6c; Supplementary Fig. 13c) for most alpha-range and all beta-range LFPs. Although this does not provide concrete proof of the causality, these analyses suggest that the alterations in the FCS network temporally led to behavioral changes, providing compelling evidence that the FCS network functionally influences behavior changes.

Finally, we examined the mechanism of how the microstimulation persistently induces negative decision-making (Supplementary Fig. 5). We focused on the *Follow-up* block, during which the behavioral effects of microstimulation persisted, but the microstimulation did not directly influence the network. We hypothesized that the signal flow in the network in the effective *Follow-up* block could exhibit a significant difference from that in the *Stim-off* block, while it does not in the non-effective session. To investigate this, we compared the DAIs between the *Stim-off* and *Follow-up* blocks in the effective sessions and found that directional influences among limbic regions were no longer significant (Fig. 6d), indicating significant reductions in DAIs between the *Stim-off* and *Follow-up* blocks (Fig. 7a). Conversely, the DAIs calculated for the non-effective sessions did not show any changes in all pairs in the FCS network (Fig. 7b). Lastly, we further tested whether the reduction of the top-down influence was observed individually in the two monkeys (S and P) for the alpha-beta frequency ranges. We confirmed dampened DAIs from dlPFC to pACC and striatum in both monkeys (Supplementary Fig. 14) and no changes in non-effective sessions (Fig. 7b). These results suggest that the attenuated signal flow in the FCS network, alongside the top-down signal originating from the dlPFC

being particularly dampened, could induce persistent negative bias observed in the effective sessions.

## Discussion

We demonstrate here a potential mechanism by which the dlPFC regulates the activity of the network of cingulo-striatal regions that have been implicated in clinical[14,49,59] and experimental studies[50,60] of depression. We microstimulated the sgACC and could experimentally produce a negative bias in the monkeys' decision-making in the Ap-Av task. When we analyzed LFPs recorded during this experimentally-induced negative decision-making, we found that the influence of the dlPFC was markedly reduced. This causal evidence for simultaneously increased negativity in decision-making and diminished dominance in causal dlPFC-limbic connectivity suggests that the reduced connectivity could be a crucial factor for the negative bias in decision-making. This mechanistic hypothesis is in accord with evidence from neuroimaging studies that have implicated the regions of the FCS network as one of the networks of regions consistently related to negative processing bias frequently reported in MDD patients[14,17]. Our findings are further in accord with evidence that the dlPFC is involved in the cognitive regulation of negative emotional response[61,62]. Our findings thus suggest that the reduced directional influence of the dlPFC on the cingulo-striatal regions could be crucial for normative affective decision-making and that disruption in this dominant prefrontal control could be a neuronal substrate of the negative bias in decision-making that is a hallmark of human MDD symptoms. We thus suggest that this prefrontal influence on a limbic system could be parallel to the top-down influence exerted by the prefrontal control of the visual system[4,33,34]. The generality of these effects across functional systems suggests that the development of powerful feedback control mediated by oscillatory rhythms could be conserved from sensory circuits to affective cortical networks and their targets.

Prior research with common marmosets suggested that over-activation of the sgACC could blunt anticipation of appetitive rewards[46,47,63]. We here, in macaques, confirmed a spatial correlation between sites behaviorally effective in producing negative bias in decision-making and cellular responses at those sites, suggesting that microstimulation of the sgACC recruited local circuits by activating neuronal networks related to avoidance behavior. In this study, we found that the number of beta oscillations encoding positive decision-related variables (i.e., positive utility) was significantly reduced by the effective microstimulation. Beta oscillations are traditionally associated with motor control[37] and attention[4,27,38]. More recently, beta oscillations have also been implicated in cognitive functions, including working memory[5], somatosensory decision-making[39], and negative bias in value judgment[35]. Diminishing beta response encoding positive utility appears comparable to the blunted anticipation to reward induced by over-activation of the sgACC reported in previous studies[46,47,63,64]. Our results indicated that the beta-band oscillatory activity, which could serve as the mediator of information flow in the FCS network, encoded positive utility and was particularly reduced by the experimentally-induced negative decision-making. Further, in our previous study[35], striatal microstimulation was observed to have no effect on beta responses encoding positive utility but instead heightened beta responses associated with negative utility. This indicated that a subset of striatal beta oscillations specifically represented negative utility, and activating the striatum could enhance the processing of negative value. Conversely, the present findings suggest that circuits influenced by sgACC activities represent beta oscillations responsive to positive utility. The activation of sgACC, in turn, appears to induce the suppression of positive values encoded by these circuits. We found the change in the encoded information of the beta responses observed broadly in the FCS network after the effective sgACC microstimulation. These results suggest that the interareal

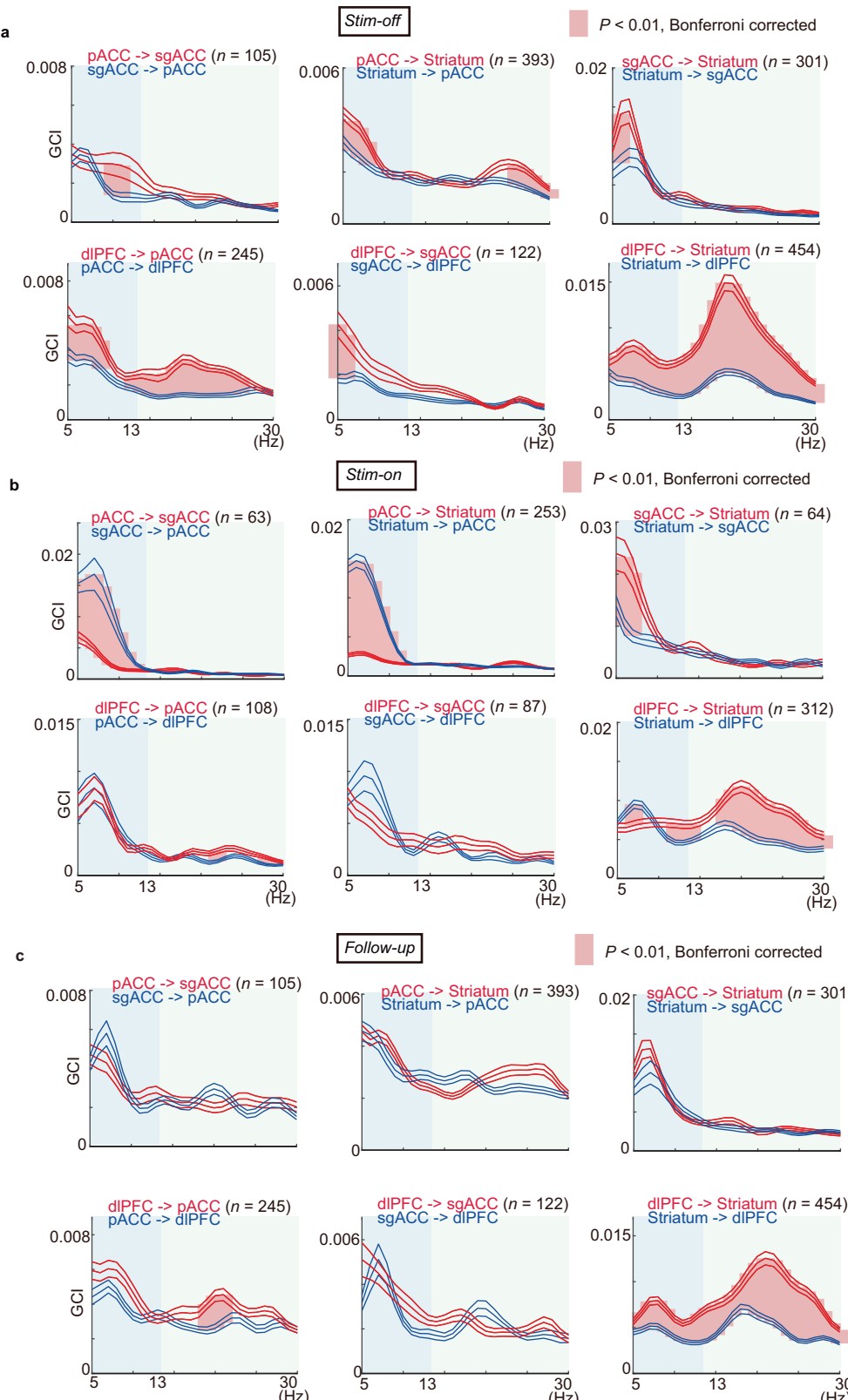

**Fig. 5 | Granger causality indices (GCIs) in alpha and beta frequency ranges (5–30 Hz) in the FCS network before, during, and after the effective micro-stimulations. a** Mean GCIs for the *Stim-off* block in the negative effective sessions. The mean GCI was plotted for each region before in alpha (blue shaded area) and beta (light gray area) frequency ranges, with the x-axis indicating frequency (Hz) and the y-axis indicating the mean GCI. The blue lines indicate the mean ± SEM of GCIs for bottom-up influences, and the red lines indicate those for the top-down influences. The thick pink lines between them indicate that the mean GCIs between the top-down and bottom-up influences were significantly different (two-sided paired t-test, *P* < 0.01, Bonferroni corrected over frequency). **b** Mean GCIs for the *Stim-on* block in the negative effective sessions. Data are presented as mean values ±SEM. **c** Mean GCIs for the *Follow-up* block in the negative effective sessions. Data are presented as mean values ±SEM. Source data are provided as a Source Data file.

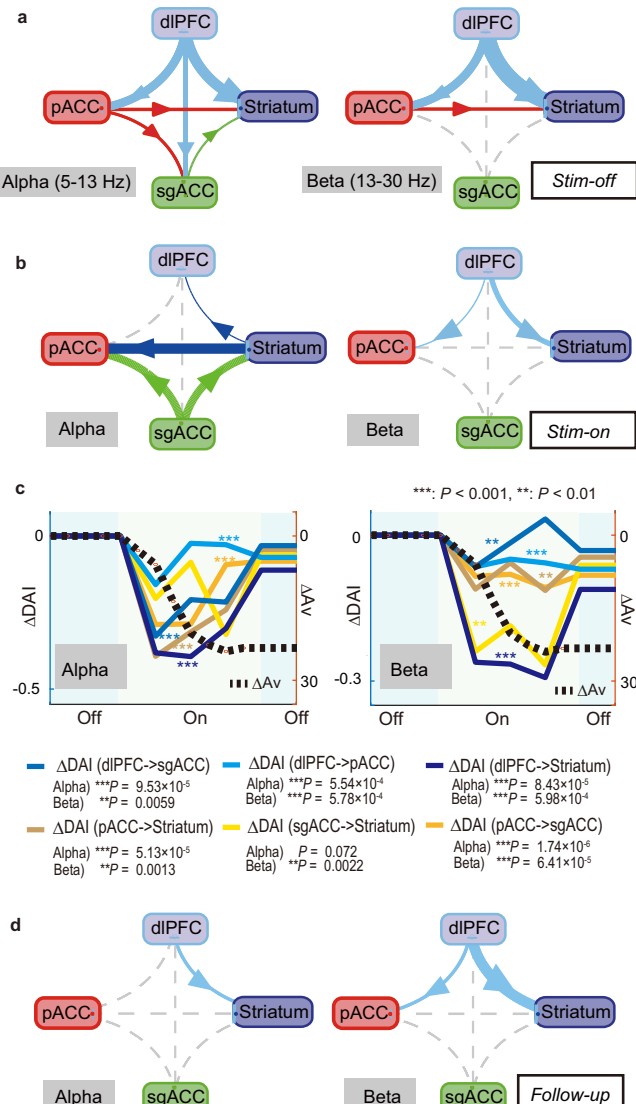

**Fig. 6 | Diagram of directional asymmetry indices (DAIs) showing the functional connectivity of the FCS network. a** The diagrams showing the Directional Asymmetry Index (DAI) in the alpha (left) and beta (right) ranges for the *Stim-off* block in the negative effective sessions. The width of the arrows indicates the magnitude of the population mean of DAIs, and the direction of the arrow indicates the direction of GCIs. The dotted line indicates that the DAI was not significantly different from zero. **b** The diagram of DAI for the *Stim-on* block in the negative effective sessions. **c** Granger causality analyses between the increase in Av (ΔAv, black dotted line) and each network change of the alpha (left) and beta (right) oscillations estimated significant causality, estimating that the network changes Granger-caused the behavior (Granger causality test, **$P < 0.01$, ***$P < 0.001$). Each color indicating each change in DAI (ΔDAI) is shown at the bottom. **d** The diagram of DAI for the *Follow-up* block in the effective sessions. Source data are provided as a Source Data file.

interactions within the large-scale FCS network could underlie the experimentally-induced negative bias in decision-making. However, we are aware that the effects that we have observed could reflect other indirect influences, which were not included in our multi-regional recordings. We did not examine, for example, the thalamus[65] or amygdala[66], each containing circuits related to emotional control. Marmoset studies have proposed that the sgACC has diverse effects on reward[64] and threat responses[45,63], which may operate through distinct pathways[46]. Prior studies suggested the multiple effects of sgACC on responses to reward and threat may act through separate pathways,

with enhanced reactivity to uncertain threat acting through the amygdala, which was not a node within the fronto-cingulo-striatal network studied here.

We estimated here how microstimulation could induce change in the interareal interaction within the FCS network. We depended on evidence that interareal synchronization can be assessed by coherence analysis of the LFPs[7,30,33], and that entrainment can be assessed by GC[7,31,32,40,67]. Recent studies of functional connectivity of widespread fronto-parietal-occipital networks[4,5,27,32,33,40] have convincingly shown that the directional influence estimated by GC is represented by the beta oscillations in these fronto-parietal[5,27], fronto-occipital[32,40] and fronto-striatal[7] networks. The direction of the signals was found to be correlated with the "top-down" direction of the anatomical cortical hierarchy[32,40–42]. Spurred by our previous finding that microstimulation-induced negative bias was associated with beta oscillations[35], we investigated the role of beta synchronization in the top-down processing of the FCS network and how microstimulation affects the directional influence of the dlPFC on the limbic regions. We observed a significant reduction of the directional influence from the dlPFC to the cingulate cortex and striatum in the FCS network. Our results thus suggest the existence of the top-down signaling from the dlPFC to the limbic system could likewise be mediated by beta synchrony, similar to that reported for the visual system, and that the blunting of such putative top-down processing could be causal to the negative bias in decision-making. We further found that alpha oscillations significantly mediated synchronous activity between the cingulate cortices and the striatum. Behaviorally effective microstimulation diminished the directional influence of the synchronous alpha activity, suggesting the importance of alpha oscillations in facilitating communication among limbic structures.

Concerning the mechanism of how sgACC microstimulation disrupts top-down control, the interplay between sgACC and dlPFC is considered critical. Previous clinical research on patients with MDD consistently reports a negative correlation in activities between cognitive and limbic regions[10]. Exploring interactions within the FCS network, the TMS on the dlPFC has been recognized for its antidepressant effects and is consistently associated with anticorrelated activities with sgACC[25]. These findings, aligning with a marmoset study[47], demonstrate that sgACC activation disrupts connectivity between sgACC and dlPFC. Our study further illustrates that the interplay between cognitive and limbic regions may be mediated by alpha and beta-range oscillations, with sgACC activation disturbing signaling within the FCS network. The sgACC microstimulation not only induced an acute change in behavior but also led to a cumulative alteration in value judgment (Supplementary Fig. 13a), which persisted in the *Follow-up* block (Supplementary Fig. 5). These findings suggest that sgACC microstimulation may influence neural plasticity mechanisms. In our previous study[51], we found that the striatal target of the cortical regions at which the microstimulation induced negative decision-making was the striosome compartment, which could potentially regulate the dopaminergic system[53]. Consequently, we hypothesize that the sgACC microstimulation might similarly influence limbic circuits involved in dopamine regulation, potentially inducing plastic changes in value judgment.

Finally, the oscillatory effects that we have found, and their hierarchical organization, could be due to non-neuronal, including glial[68] or humoral. Yet our findings raise the possibility, favored here, that in the forebrain, there is a limbic-associative parallel to the top-down processing so heavily explored for the frontal and parieto-occipital cortex. This conclusion accords well with evidence that emotion regulation[8,9], particularly suppressing negative emotions[2], is a critical function of the dlPFC[11,16,60]. Our findings provide evidence that loss of top-down influence from cognitive prefrontal to limbic system-related cortical areas could impair this suppression of negative emotion.

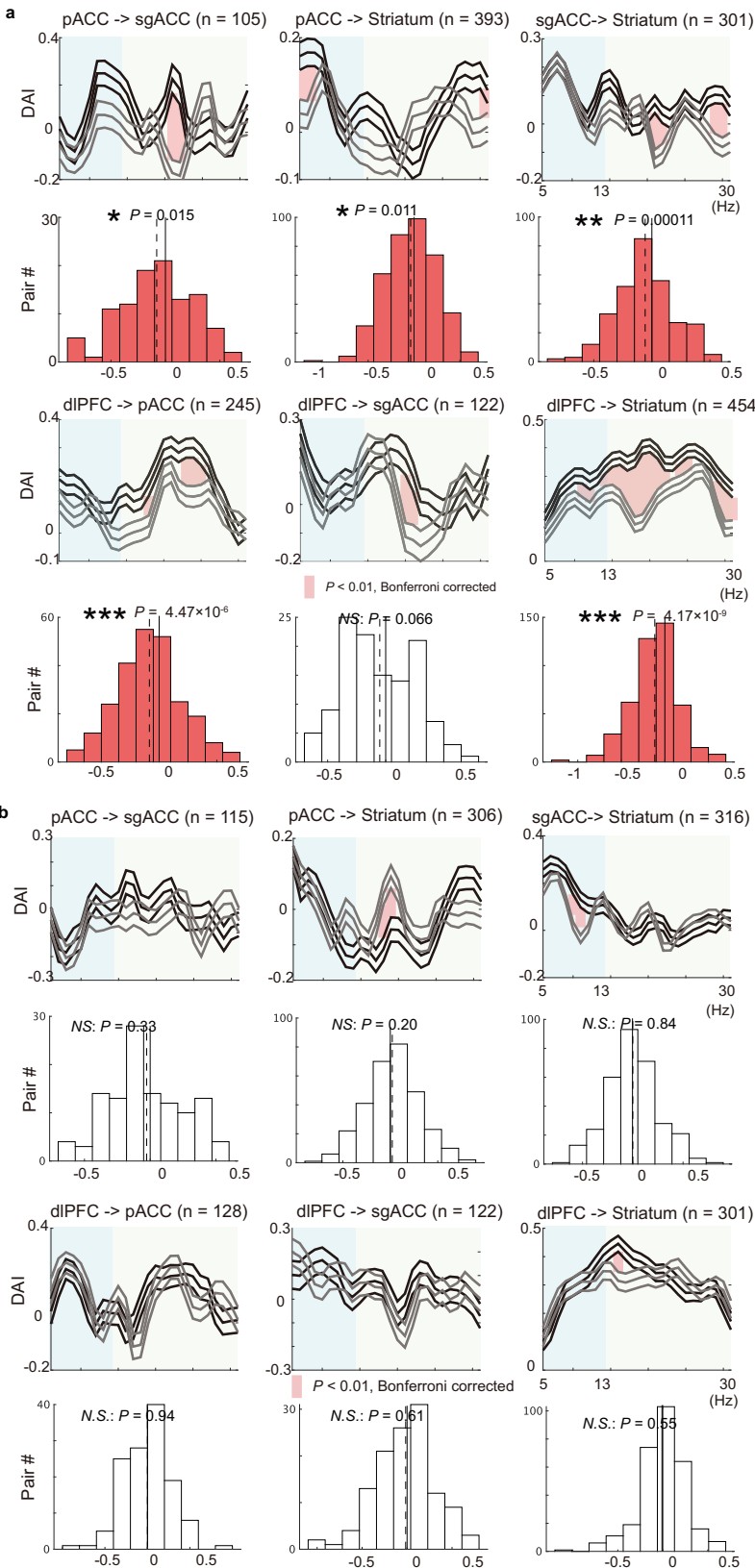

## Methods

### Subjects and task procedures

Two female *Macaca mulatta* monkeys (P, 6.3 kg; S, 7.5 kg, both 12 years old) were used in experiments conducted in accordance with the Guide for Care and Use of Laboratory Animals of the United States National Research Council and the guidelines of the Committee on Animal Care of the Massachusetts Institute of Technology (protocol # 0116-004-19). Monkeys were trained to perform approach-avoidance (Ap-Av, Fig. 1a) task[25]. When the monkey put the hand in a designated position in front of a joystick, a trial was started. The monkey was required to hold the hand to the position for 1.5 s (fixation period). After the fixation period, a visual cue consisting of red and yellow

**Fig. 7 | Change in functional connectivity was only observed in effective sessions. a** Changes in directional asymmetry index (DAI) between *Stim-off* and *Follow-up* blocks in the negative effective sessions. In the top panels, the mean DAIs, which were averaged over pairs of channels and calculated for the alpha and beta frequencies, were shown for the *Stim-off* (black) and *Follow-up* (gray) blocks. Data are presented as mean values ± SEM. The thick vertical pink lines indicate the frequencies at which the mean DAIs showed significant differences between *Stim-off* and *Follow-up* blocks ($P < 0.05$, two-sided t-test, Bonferroni corrected). The bottom panel shows the distribution of the change in mean DAI. For each pair of electrodes, the mean DAI averaged over alpha and beta frequency was calculated, and the change in the mean DAI was derived by subtracting the mean DAI for the *Stim-off* block from that for the *Follow-up* block. The vertical black dotted line indicates the mean of the distribution. The red distribution indicates significant deviation from zero (two-sided t-test, $*P < 0.05$, $**P < 0.01$, $***P < 0.001$). Most of the top-down influences (DAIs from pACC to striatum, from pACC to sgACC, from sgACC to striatum, from dlPFC to pACC, and from dlPFC to striatum) exhibited significant reduction by the effective microstimulation. **b** Change in DAIs between *Stim-off* and *Follow-up* blocks in the non-effective sessions, illustrated similarly to **a**. Top panels show the population means of DAIs for the *Stim-off* (black) and *Follow-up* (gray) blocks, calculated for each frequency. The bottom panel shows the distribution of the change in mean DAI averaged over the frequencies, showing no significant change (t-test, $P > 0.05$) induced by the non-effective stimulation. Source data are provided as a Source Data file.

horizontal bars appeared at the center of the screen. The length of the red bar linearly corresponded to the offered amount of reward (liquefied food; 0.1 ml at minimum, 2.0 ml at maximum), and the length of the yellow bar linearly corresponded to the offered strength of the airpuff (0 psi at minimum, 60 psi at maximum). Only when the monkey chose to make an approach decision these two offered outcomes were delivered. The length of the bars varied independently and pseudorandomly over 101 steps. The cues remained on for 1.5 s (decision period), and the monkey had to maintain home-position contact during this period. After the decision period, two target cues (a white square and a white cross) simultaneously appeared above and below the cue. The locations of the targets were altered randomly. At the same time, a cursor (a white opened circle) whose vertical location can be moved by the joystick is shown at the center of the screen. The choice was reported by moving a joystick to change the cursor's position toward either target within 3 s. A sound signaling "avoidance" was played when the monkey chose the square target. After 500 ms, the minimal liquefied food reward (amount of 0.1 ml and equivalent to the minimal offer by a red bar) was delivered in order to keep the monkey's motivation to perform the task. A sound signaling "approach" was played when the monkey chose the cross target. After 500 ms, an airpuff was delivered to the monkey's face for 800 ms as a pre-indicated pressure by the length of the yellow bar. The liquefied food indicated by the red bar was then delivered 1 s after the onset of the airpuff delivery for 1.5 s. A 5-s inter-trial interval was inserted between each trial. A computer-controlled pump and an air transducer controlled the amount of reward and the pressure of the airpuff.

## Procedures for control experiments

Control Experiment was conducted at one negative effective site to confirm whether the microstimulation induces eye movements or autonomic responses (Supplementary Fig. 3). One monkey was trained to perform a fixation task in which ocular fixation was required (Supplementary Fig. 3a). At the start of the task, a fixation cue (circle shape) appeared at the center of a black screen in front of the monkey. When the monkey's gaze acquired the fixation circle of 5°, a square cue appeared following the circular cue. Then, the monkey had to maintain its gaze within a fixation window for 3 s. Liquid food reward was delivered after successful fixation. Microstimulation, generated by the stimulator (Master-8, AMPI) and isolator (A365, WPI), was applied, starting at the square cue and lasting for 1 s. The stimulation induced no eye movements (Supplementary Fig. 3b, c). The microstimulation parameters were the same as those used in our microstimulation experiment in the decision period. After 100–200 trials in the no-stimulation (Stim-off) block, a block of stimulation trials (200–300 trials) followed. The pupil size was normalized by the mean size in the 3 s fixation period of the Stim-off trials before being averaged. Skin conductance (μS) measured on the monkey's palm was normalized by the value averaged over the 10 s period before the task started. For comparison, an unexpected airpuff was suddenly delivered out of the task context. We detected a significantly larger pupil size induced by

the stimulation ($P < 0.05$) (Supplementary Fig. 3d, e). The change in skin conductance caused by microstimulation was compared to that induced by a sudden airpuff (15 psi) to the monkey's face. We could detect the increase in skin conductance induced by microstimulation of the effective site. Further, we observed a significant increase in the skin conductance induced by the sudden delivery of the airpuff (Supplementary Fig. 3f, g). We thus conclude that the activation of the sgACC could cause autonomic responses.

## Recording setup

After behavioral training, a plastic recording chamber (40 × 40 mm) was implanted onto the skull by bone cement and ceramic screws at stereotaxically determined coordinates for each monkey. The sterile surgery was performed, with anesthesia induced by intramuscular ketamine (10 mg/kg) and atropine (0.05 mg/kg), followed by inhalation of 1–2.5% sevoflurane with 2 L of $O_2$. For all surgeries, analgesics were administered to the monkeys postoperatively. We injected prophylactic antibiotics on the day of surgery and daily thereafter for at least one week. A plastic grid with holes spaced at 1-mm intervals was placed onto the chamber. Magnetic resonance images (T2-weighted turbo spin echo, 300 μm in resolution, 1-mm slice thickness) were taken to identify the location of the electrode tracks that were implanted through a grid hole before and after the chamber implantation. Then, the skull overlying the targeted regions was removed with surgical anesthesia and sterile conditions. After the monkey was recovered, sets of platinum-iridium electrodes (impedance, 0.8–1.5 MΩ, FHC) were implanted into the neocortex. All electrodes were movable by custom-made micromanipulators affixed to the grid. Thirty-two electrodes were simultaneously implanted in the left hemisphere of monkey P (5 in the dlPFC, 6 in the pACC, 6 in the sgACC, 15 in the striatum). Fourty-three electrodes were implanted into the neocortex in the right hemisphere of monkey S (3 in the sgACC, 12 in the dlPFC, and 16 in the pACC), and 12 electrodes were implanted in the striatum (Fig. 1e). Five networked computers and other peripheral equipment controlled the recordings and tasks. An infrared eye camera system (Eyelink CL, SR Research) monitored the monkey's eye positions. Two computers controlled the task through a CORTEX system (National Institute of Mental Health). Task events were also sent to another personal computer that ran Matlab (MathWorks) to control the microstimulation generated by the stimulator (Master-8, AMPI) and isolator (A365, WPI). A digital data acquisition system (Digital Lynx, Neuralynx) sampled all the neural signals and the signals of task events. The Digital Lynx system amplified the neural signals collected from the microelectrodes and stored them in the hard drive. We used Offline Sorter (Plexon) to classify neural signals into single-unit activities. For the detailed analysis, we used Matlab.

## Spike, LFP recording and microstimulation

We performed 38 stimulation experiments in which we recorded spike and LFP activities from the sgACC, pACC, dlPFC, and striatum. We recorded LFPs without stimulation in 74 sessions. Activities were

recorded through eight 32-channel headstages against a silver wire implanted epidurally over the occipital cortex, which served as a common recording reference. For spike analysis, we used a band-pass filter, which ranges from 300 Hz to 9000 Hz. For LFP analysis, we used a band-pass filter, which ranges from 1 Hz to 1000 Hz. For further LFP analysis offline, the signals were re-referenced in order to remove the common recording reference.

We recorded neural activities when the monkeys performed Ap-Av or Ap-Ap tasks in alternating blocks of 150 trials (Supplementary Fig. 2a). Between blocks, we inserted a 10-s inter-block interval. During the period, a white spot that explicitly signaled the block change appeared at the center of the screen. For the microstimulation experiments, stimulation-off and stimulation-on trials were alternated in 250-trial blocks (Fig. 1a). No explicit signal was given at the block changes. The sequence of visual cues presented in the *Stim-off* block was repeated in the following *Stim-on* block. For stimulation experiments, monopolar stimulation was applied. The stimulation train consisted of 200-μs biphasic pulses, with the cathodal pulse leading to the anodal pulse. The signal was delivered at 200 Hz. The current magnitude was 80–90 μA. After the stimulation experiments were performed for all the electrodes, the electrodes were advanced simultaneously. Before starting the series of microstimulation experiments on the current electrode positions, we performed the recording-only session so that we could record LFPs from each position at once by -0.5 mm.

## Statistics & reproducibility

We performed microstimulation experiments on the two monkeys and found a significant effect of the microstimulation from them. We performed 38 stimulation experiments and 74 recording-only experiments. Recording-only experiments were conducted before each microstimulation experiment without moving the electrodes, allowing us to record LFPs from all stimulation and recording sites. No data were excluded from the analyses except for the channels with stimulation-induced artifacts.

To statistically define the effects of microstimulation, we measured the difference between decision matrices of the *Stim-off* block and the *Stim-on* block and derived the sizes of increases in the Ap choices (i.e., %ΔAp) and the Av choices (i.e., %ΔAv). We set a 5% increase as the threshold for distinguishing 'effective' from 'non-effective' sessions for each sgACC stimulation site. The change in Ap and Av decisions between the two blocks was calculated as followings. The choice in each trial was spatially smoothed with a 25-by-25 point square window in the decision matrix (Fig. 1g). After the spatial smoothing, each choice datum was stacked at each cell in the 100-by-100 point decision matrix. We then calculated the t-statistics for each point using the stacked choice data. We used Fisher's exact test to see statistically significant differences ($P < 0.05$ at each point) in the choices between two blocks. The size of the increase in Av decision between two blocks was then represented by the total size of points in the decision matrix that showed a significant increase in Av (%ΔAv). Similarly, the size of the increase in Ap decision was then represented by the size of points exhibiting a significant increase in Ap (%ΔAp). To evaluate whether the microstimulation significantly increased Av choices, we first examined the recording-only sessions to evaluate the stability in Ap-Av choices across blocks. We thus calculated the change in Av choice (i.e., %ΔAv) by comparing the first and second blocks of the Ap-Av task. These two Ap-Av blocks were separated by the Ap-Ap control task consisting of 150 trials. For each session, %ΔAv and %ΔAp were calculated (Supplementary Fig. 2a). We used record-only sessions to evaluate the misclassification rate. We set a change of 5% as a threshold to discriminate between negative effective and non-effective stimulation sessions. By the 5% threshold, record-only sessions were classified as negative effective with 2.8% (Supplementary Fig. 2b). As a false positive rate to misclassify non-effective as negative effective was

less than 5%, the threshold could thus correctly discriminate them with over 95% probability. Among 38 stimulation experiments at the sgACC, we thus defined ten sessions as negative-effective and three sessions as positive-effective sessions by this discrimination threshold. We confirmed that in the negative (positive) effective sessions, no significant increase in Ap (Av) was detected. We also adopted the saline injection experiments to evaluate the stability of the Ap-Av experiments (Supplementary Fig. 2c). In the experiments, saline was intramuscularly injected into each monkey between block 1 and block 2. The average size of the increase in the Av choice (i.e., %ΔAv) was 0.38%, and the 95% confidence limit of the mean was 1.57%. We further compared the change in the Av between the first and third blocks to estimate the confidence limit of comparing the *Stim-off* and *Follow-up* blocks. The average size of %ΔAv between the first and third blocks was 1.44%, and the 95% confidence limit of the mean was 4.05%. With these no-stimulation experiments, we concluded that with the 5% threshold, the rate of misclassification was less than 5%.

## Econometric modeling

Econometric modeling was used to analyze the probability of choosing the cross target versus the square target in the Ap-Av task. Specifically, the probability of choosing the cross target ($p_{AP}$) was calculated using the logistic function $p_{AP} = 1/(1 + \exp(-(U_{AP} - U_{AV})))$, where $U_{AP}$ and $U_{AV}$ represent the utility of each option. To approximate the function $f = U_{AP} - U_{AV}$, which captures the difference in utility between the two options, a first-order linear model was used. The model was chosen based on its Bayesian information criterion (BIC) score[25]. The function was parameterized as $f(x, y) = ax + by + c$, where $x$ and $y$ represented the length of the red and yellow bars, respectively, and a, b, and c were coefficients determined through generalized linear regression fitted to the behavioral choices. To model each utility, we used $U_{AP} = ax + by$ and $U_{AV} = -c$. The expected value of the outcome was calculated as ChV $= p_{AP} U_{AP} + (1 - p_{AP}) U_{AV}$. We defined the cost-benefit ratio (CBR) as the ratio of the sensitivities to reward and airpuff cost, which is $-b/a$. The slope of the decision boundary ($-a/b$) is the reciprocal of the CBR (Fig. 2f).

## Histological identification of electrode tracks

After removing the chamber and the grid, microelectrodes were still in place during the perfusion. The monkeys were deeply anesthetized with intravenous application of an overdose of sodium pentobarbital. The brains were perfused first by 0.9% saline, followed by 4% paraformaldehyde in 0.1 M phosphate-buffered saline (PBS). After the brains were kept in 4% paraformaldehyde for three days, all electrodes were removed. Then, the brains were blocked and stored in 25% glycerol in 0.1% sodium azide (438456; Sigma) in 0.1 M phosphate buffer (PB) at 4 °C. The blocks were frozen in dry ice on a sliding microtome and then cut into coronal sections at 40 μm. Sections were stored in 0.1% sodium azide in 0.1 M PB. To examine the electrode tracks, some sections were immunostained for glial fibrillary acidic protein (GFAP) immunofluorescence. The other sections were nissl-stained with cresylecht violet. Sections were viewed and imaged with an automatized slide scanner (TissueFAXS whole slide scanner; TissueGnostics) fitted with 10X objectives. In GFAP immunofluorescence staining, sections were rinsed three times for 2 min in 0.01 M PBS containing 0.2% Triton X-100 (Tx) (T8787; Sigma-Aldrich). Then, they were pre-treated with 3% hydrogen peroxide in PBS-Tx for 10 min. After sections were rinsed three times for 2 min in PBS-Tx, they were incubated in tyramide signal amplification blocking reagent (FP1012; PerkinElmer) in PBS-Tx (TSA block) for 60 min. Then, the sections were incubated in primary antibody solutions containing polyclonal rabbit anti-GFAP [1:500] (z0334; DAKO) in TSA block one night at 4 °C. After primary incubation, the sections were rinsed three times for 2 min in PBS-Tx and were incubated for 2 h in secondary antibody solution containing goat anti-rabbit Alexa Fluor 488 [1:300] (A11034; Invitrogen) in TSA block. After

three rinses for 2 min each in 0.1 M PB, sections were mounted onto glass slides and coverslipped with ProLong Gold antifade reagent (P36930; Invitrogen).

## Decoding of cue and outcome-period activities

To examine decision-related activity in the sgACC, we focused on the cue, airpuff, and reward periods. In the cue period, the monkeys could make a decision based on the offer indicated by visual cues, but they did not know the direction of movement required to show their decision. We determined the cue, airpuff, and reward periods as follows: the cue period is 1.5 s-period from the onset of the visual cues; the airpuff period is 1.5 s-period from the onset of the airpuff delivery; the reward period is 1.5 s-period from the onset of the reward delivery. We classified sgACC neurons with significant positive or negative person correlations with a given parameter. As for the parameters, we used Approach or Avoidance decisions for cue-period activity, the magnitude of the delivered airpuff for airpuff-period activity, and the amount of the delivered reward for reward-period activity. After the classification, we mapped the distribution of neurons related to each behavioral parameter to examine whether sub-regions contained neurons predominantly encoding specific parameters (Fig. 2b, d and Supplementary Fig. 4). We divided the recording regions into 1 mm zones into the transversely cut histological sections, pooling data from the two hemispheres for both monkeys and tested whether any sub-region contained a significantly larger proportion of a specific type of the categorized units than the other regions (Fisher's exact test, $P < 0.05$). In the sgACC, the proportions of 'avoidance neurons' and 'airpuff (+) neurons' were significantly larger in the 1 mm bin around avoidance sites (Fig. 2b, d). These results suggest a close relationship between cellular responses and pessimistic decisions.

## Beta-band LFP responses

During the experiment, we recorded the timing of the stimulation sent from the stimulator and the LFPs in each channel. To remove electrical stimulation artifacts from the raw 32 kHz-sampled files, we used linear interpolation between the time points 1.5 ms and 50 ms after the onset of the stimulation trigger pulse. However, when the recording electrode was very close to the stimulating electrode, we found that the amplifier sometimes took longer to settle into a usable range than the interval between stimulation pulses, which resulted in a distorted signal throughout the stimulation period. Therefore, we did not attempt to remove stimulation artifacts from these channels. To minimize external noise and volume-conducted neural potentials, we computed a local average reference signal by averaging the signals from all electrodes within each local electrode group. We grouped the sgACC, pACC, dlPFC, and striatal electrodes based on their anterior-posterior positions, calculated the reference signal by averaging the signals in a subregion of each group, and used the averaged LFPs of each group as the reference for each electrode in the group.

In order to reduce the file size and computational load, each channel was down-sampled by a factor of 32. This was achieved by low-pass filtering the signal in forward and reverse time with a 4th order Butterworth filter that had a cut-off frequency equal to 0.45 times the target sampling rate (1 kHz) and then selecting every 32nd sample. The signal was band-pass filtered once in forward time with a 4th-order Butterworth filter having a pass band of 13 to 28 Hz. Each filtered sample value was squared and then smoothed using a Hanning kernel 77 ms wide at half-height. The spectral analysis was done using the multitaper method, and the DC component in each time window was removed before applying tapers (Supplementary Fig. 6a, b). The window width was 1.5 s for the cue period. Spectrograms were computed using a 0.75 s window. An analytic pink noise spectrum of the form $p = af^b$ was used as a baseline to highlight small power differences. If the power spectrum of the precue period or the cue period was significantly greater than the analytic pink noise spectrum in the beta frequency band (13–28 Hz), we defined that channel as exhibiting beta oscillations. Among the 4745 LFPs recorded at least in the *Stim-off* block, 3942 LFPs (83%) had power spectra during the cue period significantly different from that of the precue period and were defined as task-related beta responses. Baseline-subtracted power spectra were calculated by subtracting the fitted pink noise spectrum from the spectrum of each task period. To examine the difference between the population spectrum of the two conditions, we employed 1-Hz bins and performed a t-test between two populations. We addressed the concerns of multiple comparisons by correcting for the number of frequency bands. We applied Bonferroni correction with $n = 25$ as we repeated the statistical analysis in the 5–30 Hz range.

## Beta response matrix and multi-dimensitonal scaling

To derive the beta response matrix, we band-pass-filtered (13–30 Hz) raw LFP activity during decision period (Fig. 3a). The magnitude of the beta oscillation (power magnitude, top right panel of Fig. 3a) was calculated by the difference between the upper and lower envelopes (blue lines in the left bottom panel of Fig. 3a). Then we averaged the power magnitude for each trial (bottom right panel of Fig. 3a) over decision period. The beta response matrix (Fig. 3c) was derived by mapping the mean power magnitude (Fig. 3b) onto the decision matrix. Multi-dimensional scaling (MDS) and clustering were performed on all beta response matrices ($n = 3716$) (Supplementary Fig. 6c, d). The correlation distance matrix between pairs of all beta responses ($D = [d_{ij}]$) was defined by $d_{ij} = 1 - r_{ij}$, where $r_{ij}$ is the cross-correlation between beta response $i$ and response $j$ recorded from each channel. The configuration matrix was derived by the MDS function (cmdscale function of MATLAB). We used top ten dimensions of the configuration matrix derived by the analyses. The dataset in the 10-dimensional space was fitted with a Gaussian mixture distribution model (fitgmdist function of MATLAB using expectation maximization algorithm; maximum iterations allowed: $10^5$; diagonal convergence type), where the optimum number of clusters was selected by BIC to be eight. The group to which each channel belonged was determined as the one with the maximum posterior probability (Supplementary Fig. 7).

## Representational similarity analysis

We conducted representational similarity analysis (RSA) to statistically compare how dlPFC, sgACC, pACC, and striatum differ in their responses to cue stimuli by utilizing the Matlab toolbox[57]. This analysis can provide insights into how the structure within the FCS network collaboratively processes information in the context of approach-avoidance decision-making. In the Ap-Av task, the lengths of the reward and the air-puff bars were continuous. Therefore, to categorize the experimental conditions, we discretized the sizes of the reward and air-puff into eight bins, resulting in 64 (=8 × 8) cue stimuli. To determine the order of experimental conditions, we decoded the utility by regressing decision patterns in the decision-making model (Supplementary Fig. 8a), and ranked the experimental conditions based on utility. Magnitudes of beta oscillation were measured while the monkeys were exposed to these 64 experimental conditions. For each brain region of interest (i.e., sgACC, striatum, pACC and dlPFC), the regional activity pattern was estimated by the population activity of beta responses recorded for each experimental condition. A dissimilarity in representation was computed for each pair of activity patterns and put into a representational dissimilarity matrix (RDM) (Supplementary Fig. 8b). The dissimilarities between the activity patterns can be considered distances in the multivariate response space, and the RDM describes the geometry of the representation, serving as a signature that can be compared between different brain regions. To visualize the relationship among the representations of four brain regions, we performed multi-dimensional scaling of the four RDMs (Supplementary Fig. 8c) and

the correlation analyses (Kendall's tau and Spearman's test) (Supplementary Fig. 8d).

### Regression analyses for beta responses

To analyze the beta responses during the cue period and determine which features they encoded (Supplementary Fig. 9), we conducted an all-possible subset regression analysis with five selected explanatory variables: offered reward size (Rew), offered airpuff size (Ave), Eutil, RT (as depicted in Fig. 1c), and frequency of omission error (FOE). Linear regression analyses were exhaustively performed using every possible combination of the five explanatory variables. We selected the combination of variables that explained the cue-period activity significantly well ($P < 0.05$, F-test of the overall fit) and produced the highest BIC score. We used other scoring techniques, such as Akaike Information Criteria (AIC), Mallow's Cp (Cp), and stepwise regression, to ensure the quality of fit. We counted the number of channels that were best explained by a single variable or a combination of variables. We confirmed that the beta responses and unit activities used in these analyses did not have multicollinearity problems, as diagnosed by Belsley's criteria. We use the term "encode" to indicate that we interpreted the unit or beta activities as exhibiting differential responses specifically to the variable. However, the explanatory variables were arbitrarily introduced, and it did not necessarily mean that the unit or beta exhibited selective responses only to these variables.

### Classification of beta responses recorded during the recording, Stim-off, Stim-on, and Follow-up blocks

In Fig. 4, we show the analysis of the beta responses recorded in the *Stim-off*, *Stim-on*, and *Follow-up* blocks. We selected channels that showed task-related beta responses in the *Stim-off* and from which we could stably record the LFPs continuously from the *Stim-off* to *Follow-up* blocks. In order to classify these 57 and 84 beta responses recorded in the three blocks of the effective and non-effective sessions, we performed MDS to derive the relative similarities among the beta responses. The number of groups assigned to the combined set was again eight (Fig. 4b).

### Coherence and Granger causality analysis

Functional connectivity between brain regions was examined by coherence and Granger causality (GC). We supposed LFPs simultaneously recorded from different brain regions as two jointly distributed vector-valued stochastic processes, $\mathbf{X} = [X_1, X_2, \ldots X_n]$, $\mathbf{Y} = [Y_1, Y_2 \ldots Y_n]$, where $n$ corresponds to the time of the cue period. We first estimated the frequency spectrum between $\mathbf{X}$ and $\mathbf{Y}$ from alpha to beta ranges (5–30 Hz) using a multitaper method (FieldTrip[69] ft_freqanalysis(), mtmfft method with tapsmofrq = 2, numtapers = 4, and NW = 3). Then, we computed coherence spectra between two LFPs of different regions by using the frequency spectrum and ft_connectivityanalysis() with cfg.method = 'coh'. The coherence does not provide information regarding the direction of information flow between brain regions. The directional influence between brain regions in alpha and beta bands was estimated by computing GC in the frequency domain. GC from $\mathbf{Y}$ to $\mathbf{X}$, written $GC_{\mathbf{Y} \to \mathbf{X}}$, stands to quantify the degree to which the past of $\mathbf{Y}$ helps predict $X$, over and above the degree to which $\mathbf{X}$ is already predicted by its own past. We computed GC influences (GCI) for the frequency domain using FieldTrip toolbox using the frequency spectrum and ft_connectivityanalysis() with cfg.method = 'granger'. Finally, we calculated the Directional Asymmetry Index (DAI) as $DAI_{\mathbf{Y} \to \mathbf{X}} = (GC_{\mathbf{Y} \to \mathbf{X}} - GC_{\mathbf{X} \to \mathbf{Y}})/(GC_{\mathbf{Y} \to \mathbf{X}} + GC_{\mathbf{X} \to \mathbf{Y}})$, as in Bastos et al.[32].

### Reporting summary

Further information on research design is available in the Nature Portfolio Reporting Summary linked to this article.

## Data availability

Source data that can be formatted in Excel were provided and have been deposited in Figshare under accession code (doi: 10.6084/m9.figshare.25679811). The Granger Causality Data generated in this study have been deposited in the Mendeley Data under accession code (doi: 10.17632/rtz7g5n8tt.2). Other data, which cannot be formatted in Excel, are available upon request via e-mail: amemori.kenichi.7s@kyoto-u.ac.jp.

## Code availability

We further uploaded the code to reproduce the analyses can be downloaded at github (https://github.com/kenamemori/natcommun24). Other codes are available upon request via the e-mail: amemori.kenichi.7s@kyoto-u.ac.jp.

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

## Acknowledgements
This research was supported by JSPS 24H02163 (K.A.), JSPS 21K19428 (K.A.), JSPS 21H05169 (K.A.), JSPS 20H03555 (K.A.), JSPS 22H04998 (K.A.), AMED 22jm0210081h0003 (K.A.), Naito Foundation (K.A. and S.A.), Takeda Science Foundation (K.A. and S.A.), NIH/NIMH P50MH119467 (A.M.G.), JSPS 21J40030 (S.A.), JSPS 21K07259 (S.A.).

## Author contributions
K.A., S.A., and A.M.G. designed the experiments and performed the surgeries. K.A. and S.A. collected the recording and stimulation data. K.A. analyzed the recording and stimulation data, with critical detailed input from S.A. and A.M.G. S.A. and A.M.G. analyzed the anatomical data. S.A., K.A., and A.M.G. wrote the manuscript; all authors edited the manuscript.

## Competing interests
The authors declare no competing interests.
