## [Peer Review File · Nature Communications]

REVIEWER COMMENTS

Reviewer #1 (Remarks to the Author):

The overall premise of this study (as stated at the end of the introduction) is to determine whether beta synchrony could subserve top down processing in the Fronto-Cingulate-Striatal circuit and whether a reduction in top down processing could result in impaired regulation of the cognitive regions.

This study is a very demanding and complex one and overall has been very well described. The results are very interesting and have the potential to provide important new insight into the circuits involved in emotion regulation and decision bias.

Whilst reading the manuscript though a number of issues were raised by the results and their interpretation that require some additional explanations. In addition, the introduction could lay out in more detail why the particular regions were chosen for study. Finally, in the abstract it is stated that “The loss of top-down directional influence from cognitive to limbic regions, we suggest, could underlie negative biases in decision-making as observed in depressive states”. However, in this particular study, dlPFC dysregulation appears to have been induced by stimulation of scACC-25 and so at some level it would have been expected that the focus would be on how activation of scACC-25 can alter connectivity between dlPFC and the rest of the FCS circuit but this aspect does not seem to have been explicitly addressed?

Specific questions are laid out below:

Introduction.

1. Overall the introduction could be somewhat clearer regards why scACC was chosen for the stimulation and why pACC was chosen as opposed to other ACC regions for recordings. In addition, what exactly is included in the fronto-cingulate-striatal network. It seems a little vague as to whether this network is loosely linked together because they include regions altered in MDD or because they are intimately connected with one another based on retrograde and anterograde tracing.

Specifically:

1a. Not clear why scACC was chosen? Have you already shown previously that stimulation in this site in macaques has this effect on negative bias or is it the first time of showing? Certainly, stimulation of scACC-25 in marmosets has already been shown to induce a negative bias in an approach avoidance task (Wallis et al, Cerebral Cortex, 2019) so it would be pertinent to make that point. Always important to highlight cross-species similarities when they occur. Related to this point the fact that increased SCR was found with scACC stimulation is also consistent with effects on cardiovascular activity in marmosets (Alexander et al, 2019, Neuron), again worth highlighting.

1b. pACC needs to be defined but I assume it signifies pregenual ACC as used in the author's other papers? Worth pointing out explicitly that this region was chosen because it had previously been shown to induce a negative affective bias on the approach-avoidance task. The diagram of electrode placements in Figure 1 is very busy and it's not easy to see where pACC is located.

1c. What about the region of the striatum chosen? Does it include projections from dIPFC, pACC and scACC? From the extended figure 1 it appears electrodes were found in the accumbens, ventromedial caudate, lateral head of the caudate and putamen? Did you find differences between these sites or were there not enough electrodes in any one site for a meaningful comparison? Justification for your rationale for targeting all these different sites should be provided.

2. The overall premise of this study is to "determine whether beta synchronisation could also subserve top down processing in the Fronto-Cingulate-Striatal circuit and that reduction in top down processing of the network could result in impaired regulation of the cognitive regions".

However, it is not clear what is meant by 'impaired regulation of the cognitive regions'? It seems in the final discussion that top-down control of the so called 'limbic system' by the dIPFC is being proposed whereas this statement in the introduction appears to imply the opposite, namely regulation of dIPFC (cognitive regions) by the limbic system. Can this be clarified please?

Results.

3. It would be helpful to the reader if the 8 different clusters of the LFPs were fully described. Currently, on page 9 it states. 'among the 8 groups the black and grey...'. This summarises the results without fully describing all 8. Related to this, in the legend to Fig 3 the 8 different colours should be defined for Figure 3F.

4. In Figure 6 it shows that the majority of directionally selective functional connections were lost following microstimulation of scACC. However to show a significant effect of microstimulation of scACC on this functional connectivity a direct comparison should be made between the 'on' and 'off'

stimulation blocks rather than as now where the statistical comparison is within the 'off' periods and 'on' periods separately. For example, it needs to be ruled out that directionality effects don't just show significance in 'stimulation off' condition but just miss significance in the 'stimulation on' period and thus when directly compared don't differ from one another.

5. It is stated on page 13 at the end of the results that "The sgACC microstimulation did not uniformly affect the FCS network, but it did consistently reduce the influence from the cognitive to the limbic regions". How selective were these effects because in Figure 6b all directional effects, not just dlPFC ones appeared to have gone, including those between pACC, sgACC and striatum?

Discussion.

6. In the second paragraph of the discussion it appears to imply that sgACC stimulation primarily induced a negative bias by reducing beta responses to positive stimuli whilst following striatal stimulation in a previous study the effect was primarily to enhance beta responses to negative stimuli. However, findings in marmosets suggest that the multiple effects of sgACC on responses to reward and threat may act through separate pathways, with enhanced reactivity to uncertain threat acting through the amygdala, which was not a node within the fronto-cingulo-striatal network studied here. A discussion that takes into account these other findings is highly relevant here. Although the amygdala is mentioned more generally as a region not included in the current network analysis in the last paragraph its specific relevance to the actions of scACC activation are not mentioned but are highly relevant.

7. Finally, I think a missing aspect of the discussion is with respect to how microstimulation of sgACC leads to a disruption of dlPFC top down control. A finding pertinent to this is the finding that pharmacological activation of scACC in marmosets reduced FDG-PET uptake in dlPFC, which mirrors somewhat the findings reported here that scACC activation disrupts dlPFC connectivity (Alexander et al, 2020, Nature Comms). The pathways by which this may occur is less than clear since scACC has very little direct connectivity with dlPFC but does perhaps with pACC and ventromedial striatum.

References.

8. On a number of occasions the references don't always match the text very well.

Ref 33, on page 4 (line 63) refers to Clarke et al which focussed on vlPFC and OFC when the text is discussing sgACC. The correct ref here should be Alexander et al 2020.

It would be more appropriate to include ref 33 (Clarke et al (PNAS 2015)) in the list of refs (27, 34 and 37) that used approach-avoidance tasks in primates, page 4, line 66.

Refs 43-45 on page 14 (line 311) are used to refer to a blunting of appetitive arousal in marmosets following scACC-25 activation but ref 46 explicitly shows this effect (Alexander et al, 2019 Neuron). In contrast refs 43-45 show the marked effects of scACC-25 on conditioned threat, anxiety and negative bias in approach-avoidance which are highly relevant to the discussion and introduction but are not explicitly referred to. Moreover, Alexander et al, 2020, Nature Comms,(ref 43) specifically shows that dlPFC is reduced using FDG-PET following scACC-25 activation which is highly pertinent to the findings of the present study.

Reviewer #2 (Remarks to the Author):

Amemori et al. conducted a thorough investigation into the fronto-cingulo-striatal network's role in the decision-making process using a combination of LFP recording and microstimulation techniques. Their study uncovers the intriguing significance of beta band responses, elucidates two distinct LFP patterns, and highlights the impact of microstimulation in inducing a negative evaluation bias. These findings not only contribute to our understanding of the top-down control influence on decision-making but also shed light on the underlying mechanisms associated with depression. The experiments conducted by the authors are well-designed to address crucial questions regarding the alterations in interareal LFPs resulting from microstimulation. While the paper offers valuable insights, there are several major comments and minor points that require further attention to enhance clarity and accessibility for readers.

1. Major comments

While the study convincingly demonstrates that microstimulation of the sgACC leads to negative evaluations by the monkeys, establishing a direct causal link between the effects of microstimulation and the observed behavioral changes remains challenging. (1) The paper would benefit from further exploration to determine conclusively whether the network-level changes induced by microstimulation were responsible for these behavioral alterations.

The authors employed a block design comprising stim-off, stim-on, and follow-up blocks. During the stim-on block, monkeys could notice discernible changes in their behaviour and possibly recognize the different blocks based on their behaviour. In some instances from prior experiments, it was observed that certain monkeys exhibited compensatory or opposing behaviours during microstimulation period, implying their awareness of their altered behaviour and an intentional effort to rectify it upon the cessation of microstimulation. (1) It might be worth considering whether the authors have explored an intermingled microstimulation schedule to address this issue. Additionally, (2) it would be valuable to discuss the authors' insights on maintaining a sustained negative bias during the follow-up block.

The authors collected 3,942 out of 4,745 LFPs for further analysis, focusing on beta power changes during the pre-cue and cue periods (referred to as Task-related LFPs). However, it is noteworthy that only Task-related LFPs are not expected to play a role in information coding and Granger causality index (GCI) at the population level. The non-task-related LFPs, which make up 803 LFPs and approximately 17% of the total LFPs, can be involved in the information coding and GCI. (1) This raises a question about the involvement of non-task-related LFPs in these processes.

Clustering analysis using PCA reveals differences in information encoding within the FCS network. However, upon closer examination of the Figure, it appears challenging to identify distinct clusters. Instead, the data seems to exhibit more of a gradient change across the PCA axes. (1) This raises the question of whether this method is suitable for determining discrete clusters or if the underlying data structure leans more toward a continuous gradient pattern.

Identifying two primary groups of LFPs, namely Positive and Negative, is crucial for behaviour decoding. Nonetheless, it's evident from Figure 3f that this classification isn't strictly dichotomous, indicating the potential existence of additional, distinct types within the dataset. This nuanced understanding of LFP categorization could have implications for a more refined interpretation of their role in behaviour. Thus, (1) it is worthwhile to analyze the connectivity in the FCS network with these groups that are not classified as P or N and to discuss their potential roles in the decision-making process.

While the experiment mainly focused on beta waves due to the absence of observed peaks, it induced curiosity about the examination of other waveforms within LFP data regarding their potential roles in information encoding and Granger causality index (GCI) analysis. (1) Have other wave types been explored in this context? This analysis is also important for assessing the extent to which the beta wave component plays a significant role in generating the negative bias observed in decision-making.

While the authors primarily concentrated on analyzing causality between regions using GCI, which is valuable for understanding temporal relationships, it is also worth exploring the similarity of information processing among these structures. (1) I propose conducting “representational similarity analysis” with the data, as outlined by Kriegeskorte and Bandettini (2008, *Frontiers in Systems Neuroscience*). This analysis can offer insights into how structures within the FCS network collectively process information in the context of decision-making.

In Figure 4b, it is evident that the beta representation of the P group decreased. However, I am curious about the behaviour of the other groups, particularly those that are not involved in the P or N categories. (1) Were there any notable changes observed in these additional groups?

In Figures 5 and 6, the focus is on highlighting unidirectional connections. However, it's well-established that loop circuits exist within cortico-striatal and cortico-cortical networks, potentially generating backward information flow. For instance, Figure 5a suggests information flow from the striatum to sgACC. (1) Have the authors conducted any analyses to assess the significance of these backward information flows?

2. Minor comments

a. In the introduction, the term 'cognitive region' is employed, but its definition appears to be more functionally oriented rather than anatomically precise. To enhance clarity, it would be beneficial for the authors to provide a more detailed explanation of the functions associated with the dorsolateral prefrontal cortex (dlPFC). This should encompass its roles in working memory, emotion regulation, and other pertinent cognitive functions to establish it as a cognitive region.

b. The increase in pupil size induced by microstimulation prompts questions about its potential implications, particularly in aspect of the salience network theory. Do the authors have any insights into the significance of this observed pupil response and its alignment with the principles of the salience network theory?

- c. It is necessary to engage in a discussion of the study's results within the broader context of the functions associated with beta waves, which have been extensively studied across various neuroscience fields. These functions encompass motor control, attention, decision-making, anxiety modulation, and facilitating communication between different brain regions. Such a discussion not only aids in understanding the potential role of the fronto-cingulo-striatal network in decision-making but also sheds light on the implications of microstimulation within this context. Integrating the findings into this comprehensive framework would provide valuable insights and enrich the interpretation of the study's results.
- d. P groups were found in most structures, whereas N groups were predominantly observed in the striatum. Do you have any insights into the potential causes of this pattern and its significance?
- e. Have you observed the suppression of beta wave prior to the movement?
- f. I would appreciate further clarification regarding the term 'non-effective sessions.' The use of a 5% discrimination threshold is a valuable criterion for defining the effectiveness of microstimulation. Does this imply that microstimulation led to decision changes in certain sessions while not eliciting such changes in others, even when applied to the same brain region?
- g. Have the authors formulated any hypotheses or insights regarding the mechanisms through which microstimulation generates a negative bias in decision-making? Is it believed that microstimulation activates or deactivates the sgPFC? Additionally, what are the expectations or predictions if the sgACC were to be inactivated in the context of the study?
- h. Page 6, line 117: It is mentioned that 25 out of 38 sites were non-effective, the term 'overall' may appear too strong.
- i. Page 5, line 101: There is a typo in the word "Recording".
- j. Page 9, line 192: There is a typo in the word "tinthe".
- k. Page 11, line 234: Should it be 'Stim-on' instead of 'Stim-off'?
- l. Page 12, line 270: It appears that 'microstimulation' should be substituted for 'a negative bias in decision-making'.
- m. Page 14, line 306: Bringing up the visuomotor system in this sentence appears to be unrelated or out of context.
- n. Page 22, line 464: There is a typo in the word "significant".

Reviewer #3 (Remarks to the Author):

Amemori S et al., Cingulate microstimulation induces negative decision-making via reduced

top-down influence on primate fronto-cingulo-striatal network

The authors report a study in which they carried out microstimulation and neurophysiology recordings while monkeys engaged in an approach avoid task. In the task, the monkeys accepted or rejected offers that combined air puff and juice rewards. Microstimulation was done in the sub-genual cingulate. Most sites induced an increase in avoidance behavior. The authors also examined evoked LFP responses in the beta band and found that an important dimension driving beta LFP responses was the approach/avoid decision. Interestingly, microstimulation decreased the number of sites at which there was a larger response in approach trials. Microstimulation also decreased the functional connectivity in the network.

This study addresses an important question about how microstimulation affects both behavior and neural activity across frontal-striatal networks. The approach avoid task is one of the few tasks that allows examination of the network underlying important decisions that may be related to depression or anxiety. Overall, this study makes an important contribution to understanding both the neural systems underlying this important decision making process, and how it may be affected by microstimulation. I do have a number of questions, however, which should be addressed.

Comments:

1. It is difficult to determine the recording and stimulation locations from Figure 1e. A figure showing the locations on coronal sections of the brain would be useful. For example, it is not clear what the pAcc site is? Also, it is important to show clearly which region of the striatum is being recorded from, given the very different connectivity of the dorsal and ventral striatum.
2. In the introduction, the circuit organization is described from a cognitive-limbic perspective, and the striatum is considered limbic. However, there are both cognitive and limbic regions in the striatum. dlPFC projects dorsally, and sgAcc projections somewhat ventrally into the striatum. Thus, the circuit organization as described in the intro is confusing. This should be clarified, and clarified in relation to the actual recording locations, as described in comment 1.
3. When within the trial was stimulation applied? This should be mentioned in the results, since this is a results first, methods last journal.
4. Increased Av could follow from more consistently choosing avoid for the avoid regions, or shifting the boundary. Which did the stimulation affect?

5. Fisher's exact test was used for Figure 1G, but this was not used for the session-level analyses, which instead used a change of 5%. How many sessions showed individually significant differences using Fisher's exact test?

6. In supplemental Fig 6d and e on the MDS analysis, what do i and j refer to? In other words, what is the distance calculated between? Trial conditions?

7. The paragraph that begins with, "To examine the autonomic responses induced by effective microstimulation, we conducted an additional experiment in which we applied microstimulation to three negative effective sites while the monkeys performed..." should perhaps be further clarified. The final sentence of the paragraph is clear, but perhaps it would be better to suggest, in the first sentence, not that you are examining autonomic responses, but you want to examine the extent to which the microstimulation drives an aversive affect state directly. It seems to have some affect, so perhaps the point should be that the effect is minor relative to the air puff?

8. In the sentence, "We defined an LFP as task-related if it showed a significant difference in beta power ($P < 0.05$, z test, Bonferroni corrected)" what is being Bonferroni corrected? Frequency bands? How many were tested?

9. In the sentence, "Among the eight groups, the black and gray groups" I assume gray should say green or dark green?

10. Again it's not clear which part of the striatum was being recorded from, but it's interesting that the striatum did not perhaps have a strong representation of the P group in the analysis in Figure 3.

11. Page 11, line 234, Stim-off should probably read stim-on.

12. It would be useful to have a more detailed discussion of how the directed coherence is calculated by the toolbox, since coherence is not direction.

13. In Fig. 6, an absence of significant interactions following stimulation does not show that the interactions were significantly difference with and without stimulation. Statistics should be done to directly test these.

Point-by-point response to the reviewers' comments

Cingulate microstimulation induces negative decision-making via reduced top-down influence on primate fronto-cingulo-striatal network.

NCOMMS-23-41647-T

Reviewer's comments are in blue.

Revised texts are in red.

Our replies are in black.

Reviewer #1 (Remarks to the Author): *The overall premise of this study (as stated at the end of the Introduction) is to determine whether beta synchrony could subserve top down processing in the Fronto-Cingulate-Striatal circuit and whether a reduction in top down processing could result in impaired regulation of the cognitive regions. This study is a very demanding and complex one and overall has been very well described. The results are very interesting and have the potential to provide important new insight into the circuits involved in emotion regulation and decision bias.*

Reply: We are pleased that the Reviewer found that our results are very interesting and have the potential to provide important new insight into the circuit involved in emotion regulation and decision bias.

Comment 1-1: Whilst reading the manuscript though a number of issues were raised by the results and their interpretation that require some additional explanations. In addition, the Introduction could lay out in more detail why the particular regions were chosen for study. Finally, in the abstract it is stated that "The loss of top-down directional influence from cognitive to limbic regions, we suggest, could underlie negative biases in decision-making as observed in depressive states". However, in this particular study, dIPFC dysregulation appears to have been induced by stimulation of scACC-25 and so at some level it would have been expected that the focus would be on how activation of scACC-25 can alter connectivity between dIPFC and the rest of the FCS circuit but this aspect does not seem to have been explicitly addressed? Specific questions are laid out below:

Reply: We sincerely appreciate the Reviewer for providing this valuable comment.

Introduction: Why are these regions chosen? Based on your suggestions, we added focus on how the potential interaction between sgACC and the FCS network. First, from the results of clinical studies on Major Depressive Disorder (MDD), we have explicitly mentioned the potential involvement of fronto-cingulate and fronto-striatal interactions in emotion regulation.

L.39: Especially, the fronto-cingulo-striatal (FCS) network, which includes the fronto-striatal circuit as the cognitive system(Chrysikou et al., 2022) and the cingulo-striatal circuit as the limbic system(Gabbay et al., 2013; Pizzagalli et al., 2001), has been implicated in so-called negative processing bias(Grimm et al., 2008; Pizzagalli & Roberts, 2022) in MDD patients(Mayberg, 2009; Xu et al., 2021), who tend to react negatively to emotionally evocative stimuli(Scheele et al., 2013). Fronto-cingulate(Drevets & Raichle, 1998) and fronto-striatal(Furman et al., 2011; Heller et al., 2013) interactions have been implicated in both cognition and emotion, highlighting the potential role of dlPFC in emotional regulation. In clinical studies, when healthy individuals confront affective challenges, an augmentation in fronto-cingulate coupling has been reported(Aizenstein et al., 2009; Margulies et al., 2007). In contrast, individuals with MDD exhibited diminished fronto-cingulate connectivity(Holmes & Pizzagalli, 2008), alongside abnormal fronto-striatal connectivity(Furman et al., 2011; Heller et al., 2013).

Regarding the Reviewer’s point about ‘how activation of scACC-25 can alter connectivity between dlPFC and the rest of the FCS circuit,’ we added an explanation of this aspect in the Introduction by explicitly stating the interaction between cognitive and limbic regions in the MDD. We stated that “the antidepressant effectiveness of dlPFC activation is linked to the anticorrelated activities of sgACC”. This highlights the importance of the interaction between sgACC and dlPFC.

L.47: Specifically, the interaction among FCS network has been explored through transcranial magnetic stimulation (TMS) on the dlPFC, commonly used for MDD treatment(Cash et al., 2021). The antidepressant effectiveness of the dlPFC activation is linked to their anticorrelated activities of the subgenual anterior cingulate cortex (sgACC)(Fox et al., 2012). However, the neuronal mechanism of how the cognitive dlPFC regulates the cingulo-striatal system has remained largely unclear.

We appreciate your feedback and are grateful for the improvement made to the Introduction based on your suggestions.

Comment 1-2: Introduction.

*1. Overall the Introduction could be somewhat clearer regards **why scACC was chosen** for the stimulation and why pACC was chosen as opposed to other ACC regions for recordings. In addition, **what exactly is included in the fronto-cingulate-striatal network**. It seems a little vague as to whether this network is loosely linked together because they include regions altered in MDD or because they are intimately connected with one another based on retrograde and anterograde tracing.*

Reply: We again thank the Reviewer for these helpful comments.

Why was sgACC chosen? We have mentioned the relationship between sgACC and MDD, and we aimed to address how the activity of sgACC influenced the behavior and the network.

L.71: The sgACC was selected as the target for microstimulation due to its critical role in emotional modulation in MDD. In MDD patients, the sgACC exhibited elevated metabolic activity, which decreased with successful antidepressant treatment (Mayberg et al., 1999). Deep brain stimulation of sgACC effectively relieved symptoms in treatment-resistant depression (Clarke et al., 2015; Mayberg et al., 2005). Consistent with the clinical observations, primate neurophysiology studies revealed that sgACC stimulation in marmosets induced a negative bias in approach-avoidance tasks (Wallis et al., 2019), accompanied by an increase in skin conductance, suggesting effects on cardiovascular activity (Alexander et al., 2020).

As for the explanation of “what exactly is included in the fronto-cingulate-striatal network,” we have stated why we chose the fronto-cingulate and fronto-striatal network. First, we have also explained that the anatomical connection between pACC and striatum of the recorded session was already confirmed by anatomy in the previous paper (Amemori et al., 2020). Further, we have confirmed that the pACC and sgACC have reciprocal connections in previous articles (Amemori et al., 2021). The fronto-striatal connectivity was also reported in previous articles (Ferry et al., 2000).

L.82: Our targets within the FCS network included the dlPFC as the cognitive area and the pACC and sgACC as cingulate areas known to have reciprocal connectivity (Amemori et al., 2021). Additionally, we targeted the dorsal part of the striatum (Amemori et al., 2018), which is recognized for receiving projections from the dlPFC (Ferry et al., 2000) and pACC (Amemori et al., 2020).

On the other hand, we do not think that the dlPFC sites that we recorded projected to the sgACC, as the Reviewer suggested. Thanks to the Reviewer’s comment, we thus have clarified that they are closely interconnected based on the forward tracing of the FCS network.

Comment 1-3: Specifically:

1a. Not clear why scACC was chosen? Have you already shown previously that stimulation in this site in macaques has this effect on negative bias or is it the first time of showing? Certainly, stimulation of scACC-25 in marmosets has already been shown to induce a negative bias in an approach avoidance task (Wallis et al, Cerebral Cortex, 2019) so it would be pertinent to make that point. Always important to highlight cross-species similarities when they occur. Related to this point the fact that increased SCR was found with scACC stimulation is also consistent with effects on cardiovascular activity in marmosets (Alexander et al,

2019, Neuron), again worth highlighting.

Reply: We apologize for this. The subgenual anterior cingulate cortex (sgACC) has been observed to be overactive in patients with depression and generalized anxiety disorder, showing promise as a target for activity modulation in deep brain stimulation. In marmosets, stimulation of sgACC is known to exhibit a negative bias in conflict tasks(Wallis et al., 2019). Building on this body of research, we have considered sgACC as a crucial node in the prefrontal network involved in pessimistic decision-making. We have addressed this in the Introduction as follows:

L.71: The sgACC was selected as the target for microstimulation due to its critical role in emotional modulation in MDD. In MDD patients, the sgACC exhibited elevated metabolic activity, which decreased with successful antidepressant treatment(Mayberg et al., 1999). Deep brain stimulation of sgACC effectively relieved symptoms in treatment-resistant depression(Clarke et al., 2015; Mayberg et al., 2005). Consistent with the clinical observations, primate neurophysiology studies revealed that sgACC stimulation in marmosets induced a negative bias in approach-avoidance tasks(Wallis et al., 2019), accompanied by an increase in skin conductance, suggesting effects on cardiovascular activity(Alexander et al., 2020).

Thank you for your insightful feedback.

Comment 1-4: 1b. pACC needs to be defined but I assume it signifies pregenual ACC as used in the author's other papers? Worth pointing out explicitly that this region was chosen because it had previously been shown to induce a negative affective bias on the approach-avoidance task. The diagram of electrode placements in Figure 1 is very busy and it's not easy to see where pACC is located.

Reply: Yes. Thank you very much for your feedback. As you pointed out, the area corresponding to the anterior cingulate cortex, where pessimistic decision-making induced by stimulation was observed in previous studies (Amemori et al., 2018), is highlighted. As you suggested, I have explicitly stated this in the following manner:

L.82: Our targets within the FCS network included the dlPFC as the cognitive area and the pACC and sgACC as cingulate areas known to have reciprocal connectivity(Amemori et al., 2021). Additionally, we targeted the dorsal part of the striatum(Amemori et al., 2018), which is recognized for receiving projections from the dlPFC(Ferry et al., 2000) and pACC(Amemori et al., 2020). Similarly to the effect of microstimulation of the pACC(Amemori & Graybiel, 2012) and striatum(Amemori et al., 2018), here we found that microstimulation of the sgACC successfully induced a negative bias in decision-making.

Moreover, we have refined the depiction of recording sites in the revised map (**Fig. 1f**) to enhance the clarity of identification for each recording region.

We sincerely value your feedback, as it has contributed to making the manuscript more comprehensible.

Comment 1-5: 1c. What about the region of the striatum chosen? Does it include projections from dlPFC, pACC and scACC? From the extended figure 1 it appears electrodes were found in the accumbens, ventromedial caudate, lateral head of the caudate and putamen? Did you find differences between these sites or were there not enough electrodes in any one site for a meaningful comparison? Justification for your rationale for targeting all these different sites should be provided.

Reply: Indeed, as the Reviewer rightly pointed out, the recorded striatal data encompasses recordings from our prior paper (Amemori et al., 2018). In our recent paper, it incorporates projections from the pACC, with a prevalence of striosomes in certain regions. The recorded area is shown in **Fig. 1f**, and it mainly includes the dorsal part of the striatum, and we did not record from the accumbens. We recorded from a part of the ventromedial caudate, a part of the head of the caudate, and the anterior putamen.

We summarized these as follows:

L.83: Additionally, we targeted the dorsal part of the striatum (Amemori et al., 2018), which is recognized for receiving projections from the dlPFC (Ferry et al., 2000) and pACC (Amemori et al., 2020). Similarly to the effect of microstimulation of the pACC (Amemori & Graybiel, 2012) and striatum (Amemori et al., 2018), here we found that microstimulation of the sgACC successfully induced a negative bias in decision-making.

As for “differences between these sites,” we did not thoroughly map the striatal beta oscillation, but we have already reported the detail of the features of the beta responses (Amemori et al., 2018 Neuron; Amemori et al., Frontier 2020). So, we did not restate in the current manuscript.

Comment 1-6:

2. *The overall premise of this study is to “determine whether beta synchronisation could also subserve top down processing in the Fronto-Cingulate-Striatal circuit and that reduction in top down processing of the network could result in impaired regulation of the cognitive regions”. However, it is not clear what is meant by ‘impaired regulation of the cognitive regions’? It seems in the final Discussion that top-down control of the so called ‘limbic system’ by the dlPFC is being proposed whereas this statement in **the Introduction** appears to imply the opposite, namely regulation of dlPFC (cognitive regions) by the limbic system. Can this be clarified please?*

Reply: We again thank the Reviewer for these helpful comments. In the revised Introduction, we highlighted the importance of the interaction between sgACC and dlPFC.

L.47: Specifically, the interaction among FCS network has been explored through transcranial magnetic stimulation (TMS) on the dlPFC, commonly used for MDD treatment(Cash et al., 2021). The antidepressant effectiveness of the dlPFC activation is linked to their anticorrelated activities of the subgenual anterior cingulate cortex (sgACC)(Fox et al., 2012). However, the neuronal mechanism of how the cognitive dlPFC regulates the cingulo-striatal system has remained largely unclear.

Further, in the revised Introduction, we explicitly mentioned that:

L.88: These findings suggest that the activity of the cognitive dlPFC has a crucial function in regulating negative emotional bias processed by the limbic system.

The statement is consistent with the statement in the Discussion part. Thank you for your insightful feedback.

Comment 1-7:

Results.

3. *It would be helpful to the reader if the 8 different clusters of the LFPs were fully described. Currently, on page 9 it states. ‘among the 8 groups the black and grey....’. This summarises the results without fully describing all 8. Related to this, in the legend to Fig 3 the 8 different colours should be defined for Figure 3F.*

Reply: We thank the Reviewer for pointing out this important point. In the revised **Fig. 3d**, we added the population activities of other groups. We explained that the green and blue groups had high magnitude along with the decision boundaries, while cyan group activated for the low aversive condition. Yellow

groups showed activation for low and high reward conditions.

L.206: Among the eight groups, the black and gray groups had the lowest and the second lowest PCVs (**Fig. 3d**) and were categorized as P (positive) group. The pink and red groups had the highest and the second highest PCVs and were categorized as the N (negative) group. The clustering further demonstrated distinct groups of LFPs, indicating specific activations for the low airpuff offer (cyan group in **Fig. 3d**), along with other groups showing activations for the decision boundary (blue and green groups in **Fig. 3d**) and that showing activity for low and high reward offers (yellow group in **Fig. 3d**).

Thank you for your insightful feedback.

Comment 1-8: 4. In Figure 6 it shows that the majority of directionally selective functional connections were lost following microstimulation of scACC. However to show a significant effect of microstimulation of scACC on this functional connectivity, a direct comparison should be made between the ‘on’ and ‘off’ stimulation blocks rather than as now where the statistical comparison is within the ‘off’ periods and ‘on’ periods separately. For example, it needs to be ruled out that directionality effects don’t just show significance in ‘stimulation off’ condition but just miss significance in the ‘stimulation on’ period and thus when directly compared don’t differ from one another.

Reply: We value the Reviewer’s comment. Initially, our intention was to compare the two distinct *Stim-off* periods before and after the effective stimulation. This approach aimed to mitigate the potential influence of stimulation artifacts when analyzing the LFPs during the Stim-on period. In the revised manuscript, we meticulously examined the LFPs and ensured the successful removal of stimulation artifacts using the method outlined in the Methods section.

L.629: To remove electrical stimulation artifacts from the raw 32 kHz-sampled files, we used linear interpolation between the time points 50 ms and 1.5 ms after the onset of the stimulation trigger pulse. However, when the recording electrode was very close to the stimulating electrode, we found that the

amplifier sometimes took longer to settle into a usable range than the interval between stimulation pulses, which resulted in a distorted signal throughout the stimulation period.

After eliminating the artifact caused by the stimulation, we conducted GCI directionality analyses. Remarkably, microstimulation significantly reduced the ‘top-down’ influence in the beta frequency range while concurrently enhancing the ‘bottom-up’ influence mediated by alpha waves in the *Stim-on* block. From the GCIs in the *Stim-on* block, we derived the directional influences of the signals, illustrated in **Fig. 6b** with diagrams depicting the Directional Asymmetry Index (DAI). During stimulation, there was a significant attenuation of top-down signals through beta waves, and concurrently, alpha waves exhibited a strengthening of bottom-up signals. This comparison directly confirmed the reduction in beta waves due to stimulation.

We described these findings in the main text as:

L.310: We next tested the prediction that sgACC microstimulation could attenuate top-down influences from the dlPFC. To calculate directional influences in the network, we compared the difference of GCIs across pairs of LFPs (**Fig. 6b, c**). The microstimulation significantly reduced the top-down influence in the beta oscillation while enhancing the bottom-up influence mediated by alpha oscillation in the FCS network. Specifically, in the alpha range, GCIs from the sgACC and striatum indicated a pronounced strengthening of the bottom-up influences. In the beta range, the top-down influences originating from the dlPFC were significantly attenuated (**Fig. 6b**).

We appreciate the Reviewer’s insightful comment, which drastically improved the quality of the manuscript.

Comment 1-9: 5. It is stated on page 13 at the end of the results that “The sgACC microstimulation did not uniformly affect the FCS network, but it did consistently reduce the influence from the cognitive to the limbic regions”. How selective were these effects because in Figure 6b all directional effects, not just dlPFC ones appeared to have gone, including those between pACC, sgACC and striatum?

Reply: Thank you so much for pointing this out. In this section, we meant to confirm that the reduction of the top-down signal is consistently observed in two monkeys independently (**Fig. 7b** and **Supplementary Figure 14**), which led us to write “consistent effect.” Following your suggestion, we explicitly stated these features in the main text:

L.326: Finally, we examined the mechanism of how the microstimulation persistently induces negative decision-making (**Supplementary Fig. 5**). We focused on the *Follow-up* block, during which the behavioral effects of microstimulation persisted, but the microstimulation did not directly influence the network. We hypothesized that the signal flow in the network in the effective *Follow-up* block could exhibit a significant difference from that in the *Stim-off* block, while it does not in the non-effective session. To investigate this, we compared the DAIs between the *Stim-off* and *Follow-up* blocks in the effective sessions and found that directional influences among limbic regions were no longer significant (**Fig. 6d**), indicating significant reductions in DAIs between the *Stim-off* and *Follow-up* blocks (**Fig. 7a**). Conversely, the DAIs calculated for the non-effective sessions did not show any changes in all pairs in the FCS network (**Fig. 7b**).

In the last part of this paragraph, we interpreted these results as:

L.335: Lastly, we further tested whether the reduction of the top-down influence was observed individually in the two monkeys (S and P) for the alpha-beta frequency ranges. We confirmed dampened DAIs from dlPFC to pACC and striatum in both monkeys (**Supplementary Fig. 14**) and no changes in non-effective sessions (**Fig. 7b**). These results suggest that the attenuated signal flow in the FCS network, alongside the top-down signal originating from the dlPFC being particularly dampened, could induce persistent negative bias observed in the effective sessions.

We apologize the confusion in the previous manuscript and appreciate the Reviewer’s corrections.

Comment 1-10: Discussion.

6. In the second paragraph of the Discussion it appears to imply that sgACC stimulation primarily induced a negative bias by reducing beta responses to positive stimuli whilst following striatal stimulation in a previous study the effect was primarily to enhance beta responses to negative stimuli. However, findings in marmosets suggest that the multiple effects of sgACC on responses to reward and threat may act through separate pathways, with enhanced reactivity to uncertain threat acting through the amygdala, which was not a node within the fronto-cingulo-striatal network studied here. A discussion that takes into account these other findings is highly relevant here. Although the amygdala is mentioned more generally as a region not included in the current network analysis in the last paragraph it’s specific relevance to the actions of scACC activation are not mentioned but are highly relevant.

Reply: Thank you for this comment. We are aware of the critical involvement of the amygdala in this process, and explicitly mentioned. In the last paragraph of Discussion, we noted:

L.384: However, we are aware that the effects that we have observed could reflect other indirect influences, which were not included in our multi-regional recordings. We did not examine, for example, the thalamus(Zikopoulos & Barbas, 2012) or amygdala(Etkin et al., 2006), each containing circuits related to emotional control. Marmoset studies have proposed that the sgACC has diverse effects on reward(Alexander et al., 2019) and threat responses(Clarke et al., 2015; Wallis et al., 2017), which may operate through distinct pathways(Wallis et al., 2019). Prior studies suggested the multiple effects of sgACC on responses to reward and threat may act through separate pathways, with enhanced reactivity to uncertain threat acting through the amygdala, which was not a node within the fronto-cingulo-striatal network studied here.

Thank you so much for your comment. We think the manuscript is much improved.

*Comment 1-11: 7. Finally, I think a missing aspect of the Discussion is with respect to **how microstimulation of sgACC leads to a disruption of dlPFC top down control**. A finding pertinent to this is the finding that pharmacological activation of scACC in marmosets reduced FDG-PET uptake in dlPFC, which mirrors somewhat the findings reported here that scACC activation disrupts dlPFC connectivity (Alexander et al, 2020, Nature Comms). The pathways by which this may occur is less than clear since scACC has very little direct connectivity with dlPFC but does perhaps with pACC and ventromedial striatum.*

Reply: We very much appreciate your comment and added further Discussion as follows:

L.410: Concerning the mechanism of how sgACC microstimulation disrupts top-down control, the interplay between sgACC and dlPFC is considered critical. Previous clinical research on patients with MDD consistently reports a negative correlation in activities between cognitive and limbic regions(Cash et al., 2021). Exploring interactions within the FCS network, the TMS on the dlPFC has been recognized for its antidepressant effects and is consistently associated with anticorrelated activities with sgACC(Fox et al., 2012). These findings, aligning with a marmoset study(Alexander et al., 2020), demonstrate that sgACC activation disrupts connectivity between sgACC and dlPFC. Our study further illustrates that the interplay between cognitive and limbic regions may be mediated by alpha and beta-range oscillations, with sgACC activation disturbing signaling within the FCS network.

Thank you so much for your comment. We think the manuscript is much improved.

Comment 1-12: References.

8. On a number of occasions the references don't always match the text very well. Ref 33, on page 4 (line 63) refers to Clarke et al which focussed on vIPFC and OFC when the text is discussing sgACC. The correct ref here should be Alexander et al 2020. It would be more appropriate to include ref 33 (Clarke et al (PNAS 2015)) in the list of refs (27, 34 and 37) that used approach-avoidance tasks in primates, page 4, line 66. Refs 43-45 on page 14 (line 311) are used to refer to a blunting of appetitive arousal in marmosets following scACC-25 activation but ref 46 explicitly shows this effect (Alexander et al, 2019 Neuron). In contrast refs 43-45 show the marked effects of scACC-25 on conditioned threat, anxiety and negative bias in approach-avoidance which are highly relevant to the Discussion and Introduction but are not explicitly referred to. Moreover, Alexander et al, 2020, Nature Comms,(ref 43) specifically shows that dlPFC is reduced using FDG-PET following scACC-25 activation which is highly pertinent to the findings of the present study.

Reply: We thank the Reviewer for raising this important point. We cited Alexander et al 2020 in Introduction part that mentioned the sgACC function:

L.74: Consistent with the clinical observations, primate neurophysiology studies revealed that sgACC stimulation in marmosets induced a negative bias in approach-avoidance tasks(Wallis et al., 2019), accompanied by an increase in skin conductance, suggesting effects on cardiovascular activity(Alexander et al., 2020).

We cited Clarke et al (2015) at the part we mentioned the Ap-Av task used in non-human primates:

L.77: Based on these findings, we conducted microstimulation on the sgACC, while macaque monkeys performed the approach-avoidance conflict task(Amemori et al., 2015; Amemori & Graybiel, 2012), a task that has been used to quantify how negative bias processing affects decision-making in both humans(Ironside et al., 2020) and non-human primates(Amemori et al., 2018; Amemori & Graybiel, 2012; Amemori et al., 2020; Clarke et al., 2015).

Moreover, we added (Alexander et al, 2019 Neuron) to the place where we mentioned the blunted reward anticipation:

L.370: Diminishing beta responses encoding positive utility appears comparable to the blunted anticipation to reward induced by over-activation of the sgACC reported in previous studies(Alexander et al., 2019; Alexander et al., 2020; Wallis et al., 2017; Wallis et al., 2019).

Further, we added (Alexander et al., 2020, Nature Comms) to the place where we mentioned anticorrelated activity between sgACC and dlPFC:

L.414: These findings, aligning with a marmoset study(Alexander et al., 2020), demonstrate that sgACC activation disrupts connectivity between sgACC and dlPFC.

Thank you so much for your valuable and precise comments.

Reviewer #2:

Reviewer #2 (Remarks to the Author):

Amemori et al. conducted a thorough investigation into the fronto-cingulo-striatal network's role in the decision-making process using a combination of LFP recording and microstimulation techniques. Their study uncovers the intriguing significance of beta band responses, elucidates two distinct LFP patterns, and highlights the impact of microstimulation in inducing a negative evaluation bias. These findings not only contribute to our understanding of the top-down control influence on decision-making but also shed light on the underlying mechanisms associated with depression. The experiments conducted by the authors are well-designed to address crucial questions regarding the alterations in interareal LFPs resulting from microstimulation. While the paper offers valuable insights, there are several major comments and minor points that require further attention to enhance clarity and accessibility for readers.

Reply: We are grateful to this Reviewer for considering our results potentially important, and we thank the Reviewer for his/her thoughtful comments. We hope that below we have replied adequately to all of the Reviewer's concerns, including that we have clearly stated now that these data do not prove causally that the oscillatory activity generates decision-making changes. Specifically, the causal-mechanistic role was answered in the replies from *Comment 2-1* to *Comment 2-6*.

Comment 2-1: 1. Major comments

While the study convincingly demonstrates that microstimulation of the sgACC leads to negative evaluations by the monkeys, establishing a direct causal link between the effects of microstimulation and the observed behavioral changes remains challenging. (1) The paper would benefit from further exploration to determine conclusively whether the network-level changes induced by microstimulation were responsible for these behavioral alterations.

Reply: We appreciate the insightful feedback from the Reviewer. In response to this comment, we conducted a thorough analysis of the temporal dynamics of stimulation effects. Specifically, we compared the onset of behavioral changes with that of the changes in the FCS network. With this analysis, we could conclude that the change in the FCS network could Granger-cause the change in behavior. First, we observed that the behavioral changes induced by the stimulation exhibited temporary accumulation of the stimulation effect, with their significance peaking in the latter stages of the stimulation block (**Supplementary Fig. 13a**). These features were also reported in our previous studies of pACC and CN microstimulation (Amemori et al., 2012 and 2017) and were robustly observed in these microstimulation experiments.

Conversely, alterations in the FCS network started in the initial phase of the stimulation block (**Supplementary Fig. 13c**). For each combination of the brain regions, we calculated the granger causality between the change in the network direction (Δ DAI) and the change in behavior (Δ Av) and found that Δ DAI “granger-causes” Δ Av for most of the alpha-range LFPs and all the beta-range LFPs of the combinations.

These findings indicate that the change in the FCS network temporarily led to behavioral changes, giving convincing evidence that the FCS network functionally influences the behavior changes.

We could thus conclude that the changes in the FCS network observed prior to the emergence of behavioral changes may underlie the mechanistic basis for the negative bias in decision-making. We showed in the main Figure (**Fig. 6c**) and summarized these results as:

L.317: We further explored the causal relationship between the network-level changes and the behavioral alterations. Notably, the behavioral changes induced by the stimulation exhibited temporal accumulation (**Supplementary Fig. 13a**), a feature consistently reported in our previous studies (Amemori et al., 2018; Amemori & Graybiel, 2012). Importantly, the changes in the FCS network began earlier than the increase in avoidance choices (**Supplementary Fig. 13b**). Granger causality analyses demonstrated that the network changes Granger-cause the increase in avoidance choices (**Fig. 6c**; **Supplementary Fig. 13c**) for most alpha-range and all beta-range LFPs. Although this does not provide concrete proof of the causality, these analyses suggest that the alterations in the FCS network temporally led to behavioral changes, providing compelling evidence that the FCS network functionally influences behavior changes.

Thank you so much for your comment. We think the manuscript is much improved.

Comment 2-2:

The authors employed a block design comprising stim-off, stim-on, and follow-up blocks. During the stim-on block, monkeys could notice discernible changes in their behaviour and possibly recognize the different blocks based on their behaviour. In some instances from prior experiments, it was observed that certain monkeys exhibited compensatory or opposing behaviours during microstimulation period, implying their awareness of their altered behaviour and an intentional effort to rectify it upon the cessation of microstimulation. (1) It might be worth considering whether the authors have explored an intermingled microstimulation schedule to address this issue.

Reply: We sincerely appreciate the insightful feedback provided by the Reviewer. Our initial observation revealed that the behavioral changes induced by the stimulation displayed a temporary accumulation of the stimulation effect, reaching peak significance in the latter stages of the stimulation block (**Supplementary Fig. 13a**). Notably, similar patterns were identified in our prior studies involving pACC (Amemori & Graybiel, 2012) (Amemori & Graybiel, 2012) and striatal microstimulation (Amemori et al., 2018) (Amemori et al., 2012 and 2017). These robustly observed features consistently manifested in the context of microstimulation experiments.

We described these features in the main text:

L.317. We further explored the causal relationship between the network-level changes and the behavioral alterations. Notably, the behavioral changes induced by the stimulation exhibited temporal accumulation (**Supplementary Fig. 13a**), a feature consistently reported in our previous studies (Amemori et al., 2018; Amemori & Graybiel, 2012).

These results suggest that each microstimulation of the sgACC gradually modulated the weighing between reward and punishment. However, it raised the possibility that the microstimulation did not influence the Ap-Av decision-making itself.

As wisely noted by the Reviewer, it is crucial to assess the direct impact of microstimulation on Ap-Av decision-making through “*an intermingled microstimulation schedule*”. Notably, in our prior manuscript, we employed a randomized stimulation approach, comparing Ap-Av decisions between *Stim-on* and *Stim-off* trials in a randomized order rather than in blocks ($n = 6$). No substantial changes in Ap-Av decisions were observed. Instead, a noticeable alteration in the slope of the decision boundary was evident, transitioning from a yellow dotted to a white dotted line, as illustrated in the Figure below.

Figure 7d of (Amemori & Graybiel, 2012). Lack of stimulation-induced difference in experiments with randomly presented stimulation-on trials (white dotted line) and stimulation-off trials (black dotted line). Yellow dotted line indicates decision boundary of collected data obtained in the stimulation-off trials in previous sessions.

Based on these previous findings, we consider that microstimulation could induce plastic change in value judgment. We added the following Discussion in the **Supplementary Discussion**:

L.241 of Supplementary Information. The effect of microstimulation on the decision-making process is characterized by an accumulative change in value judgment. In analyzing ten effective sessions, we segmented the *Stim-on* block into six temporal periods (*Stim-on1*, ..., *Stim-on6*). Choice data for each period

were aggregated across the ten effective sessions following the standardization procedure (see **Methods**). We assessed the difference in choice patterns between *Stim-off* and each of the six temporal periods. Notably, we observed a temporal accumulation of the stimulation effect on behavioral changes, reaching peak significance in the later stages of the stimulation block (**Supplementary Fig. 13a**). These characteristics align with findings from our prior studies on pACC (Amemori & Graybiel, 2012) and striatal (Amemori et al., 2018) microstimulation, indicating a robust pattern across various microstimulation experiments. The results imply a gradual modulation of reward-punishment weighting by sgACC microstimulation, with no direct influence on Ap-Av decisions. Examining the direct influence on the Ap-Av decision-making required a randomized order of stimulation on and off trials. Our previous work (Amemori & Graybiel, 2012), involved such randomization, revealing no substantial changes in Ap-Av decisions. However, we did observe alterations in the slope of the decision boundary instead (Figure 7d of (Amemori & Graybiel, 2012)). Based on these results, we concluded that sgACC microstimulation did not influence the decision-making process but might influence the neural plasticity of the value judgment.

Additionally, (2) it would be valuable to discuss the authors' insights on maintaining a sustained negative bias during the follow-up block.

Reply: We thank the Reviewer to giving us the opportunity to discuss the mechanism of this “persistent” effect of the microstimulation. In Discussion, we added:

L.418. The sgACC microstimulation not only induced an acute change in behavior but also led to a cumulative alteration in value judgment (**Supplementary Fig. 13a**), which persisted in the *Follow-up* block (**Supplementary Fig. 5**). These findings suggest that sgACC microstimulation may influence neural plasticity mechanisms. In our previous study (Amemori et al., 2020), we found that the striatal target of the cortical regions at which the microstimulation induced negative decision-making was the striosome compartment, which could potentially regulate the dopaminergic system (Amemori et al., 2021). Consequently, we hypothesize that the sgACC microstimulation might similarly influence limbic circuits involved in dopamine regulation, potentially inducing plastic changes in value judgment.

We appreciate the Reviewer for providing us with the opportunity to clarify and discuss these issues.

Comment 2-3:

The authors collected 3,942 out of 4,745 LFPs for further analysis, focusing on beta power changes during the pre-cue and cue periods (referred to as Task-related LFPs). However, it is noteworthy that only Task-related LFPs are not expected to play a role in information coding and Granger causality index (GCI) at the population level. The non-task-related LFPs, which make up 803 LFPs and approximately 17% of the

total LFPs, can be involved in the information coding and GCI. (1) This raises a question about the involvement of non-task-related LFPs in these processes.

Reply: We appreciate the valuable feedback from the Reviewer. In response to their concerns, we conducted clustering, containing 803 non-task-related LFPs. Despite being defined by previous criteria, these non-task-related LFPs still exhibited features related to the Ap-Av decision-making process. Therefore, in our new analysis, we decided to include these 803 LFPs. Instead, we implemented a refined criterion to ensure data quality. We selected LFPs that reliably generated the 'beta response matrix' for Multidimensional Scaling (MDS) analyses and could also contribute to produce the 'temporal pattern of decision-related beta responses' as illustrated in **Fig. 3d**. Some LFPs that may contain strong (potential) artifact were then not used. Subsequently, 3716 LFPs, inclusive of both task-related and non-task-related LFPs, were used for subsequent analyses. We then omitted the old definition of task-related LFPs, updating the Results accordingly:

L.186. We analyzed a total of 3,716 LFPs recorded during the recording sessions or the Stim-off block of the microstimulation experiments. Specifically, 188 were recorded from the sgACC, 1,696 from the striatum, 680 from the pACC, and 1,152 from the dlPFC, all of which were utilized in the subsequent analyses.

In the Granger analyses (**Figures 5-7**), we used non-task-related LFPs without discriminating based on the feature of the beta spectrum. We truly appreciate the Reviewer's keen insight on the LFP analyses.

Comment 2-4:

Clustering analysis using PCA reveals differences in information encoding within the FCS network. However, upon closer examination of the Figure, it appears challenging to identify distinct clusters. Instead, the data seems to exhibit more of a gradient change across the PCA axes. (1) This raises the question of whether this method is suitable for determining discrete clusters or if the underlying data structure leans more toward a continuous gradient pattern.

Reply:

We appreciate the thoughtful feedback from the Reviewer. Upon careful consideration of the dataset, we acknowledge the apparent continuity rather than distinct clusters. To address this, we improve the clustering method. When we performed the clustering, we used a high-dimensional dataset. We expanded the dimensionality of the data from 4 to 10 during clustering, and the results were then projected onto a two-dimensional MDS space. This parameter adjustment significantly enhanced the classification quality, particularly for distinguishing between the P and N groups.

In this refined analysis, we computed posterior probabilities for each datum. The analysis provided insights into how well each group could predict the indication of a given data point. Notably, the two-dimensional MDS map exhibited a clearer separation between the P and N groups. Specifically, when calculating the posterior probabilities for data points belonging to the black and gray P groups, 99.6% of black data had posterior probabilities of 75% or higher for the P group, and 72.9% of gray data had posterior probabilities exceeding 75% for the P group. Similarly, minimal overlap with other data was observed for the N group on the two-dimensional MDS map.

Furthermore, detailed statistics were provided for each group, highlighting the success of the clustering. For 79.7% of the P group, the posterior probability of being categorized as part of the P group exceeded 75%, and for 76.1% of the N group, the posterior probability of being classified as part of the N group was over 75%. This substantiates the efficacy of the clustering process for a significant portion of the data classified into N and P groups. We described these features in the Results as:

L.212: Acknowledging the continuity rather than the distinct separation between similar clusters, we quantified predictive accuracy using posterior probabilities. Remarkably, 79.7% of the P group and 76.1% of the N group had posterior probabilities exceeding 75% (Supplementary Fig. 7a), suggesting that the clustering process was efficient for a substantial portion of the data classified into N and P groups.

However, as wisely noted by the Reviewer, the posterior probabilities for the remaining groups (green, yellow, blue, and cyan) did not consistently surpass 75%, indicating the presence of gradient patterns within the dataset. We thus explicitly presented the degree of successful clustering for each group by showing the posterior probability for individual data points in **Supplementary Fig. 7**.

Comment 2-5:

Identifying two primary groups of LFPs, namely Positive and Negative, is crucial for behaviour decoding. Nonetheless, it's evident from Figure 3f that this classification isn't strictly dichotomous, indicating the potential existence of additional, distinct types within the dataset. This nuanced understanding of LFP categorization could have implications for a more refined interpretation of their role in behaviour. Thus, (1) it is worthwhile to analyze the connectivity in the FCS network with these groups that are not classified as P or N and to discuss their potential roles in the decision-making process.

Reply: We thank the Reviewer for the comment. In the light of this valuable comment, we examined how the effective microstimulation changed the representation of the groups, which were not classified as N and P groups, and presented them in new **Fig. 3d**. We explained that the green and blue groups had high magnitude along with the decision boundaries, while cyan group activated for the low aversive condition. Yellow groups showed activation for low and high reward conditions.

L.206: Among the eight groups, the black and gray groups had the lowest and the second lowest PCVs (**Fig. 3d**) and were categorized as P (positive) group. The pink and red groups had the highest and the second highest PCVs and were categorized as the N (negative) group. The clustering further demonstrated distinct groups of LFPs, indicating specific activations for the low airpuff offer (cyan group in **Fig. 3d**), along with other groups showing activations for the decision boundary (blue and green groups in **Fig. 3d**) and that showing activity for low and high reward offers (yellow group in **Fig. 3d**).

Thank you for your insightful feedback. As for the examining the FCS network, all the LFPs simultaneously recorded during the stimulation experiments were used. The connectivity in the FCS network were examined and discussed without using the classification of P and N groups.

Comment 2-6:

While the experiment mainly focused on beta waves due to the absence of observed peaks, it induced curiosity about the examination of other waveforms within LFP data regarding their potential roles in information encoding and Granger causality index (GCI) analysis. (1) Have other wave types been explored in this context? This analysis is also important for assessing the extent to which the beta wave component plays a significant role in generating the negative bias observed in decision-making.

Reply: We thank the Reviewer for this insightful comment. We do agree that we should think of the possibility of the function of other waveforms. In short, we separated the alpha and beta components in the

Granger analyses and confirmed differential changes induced by the microstimulation. In **Supplementary Fig. 6**, we have analyzed the existence of peaks in the power spectrum and found that most of the peaks existed in the beta ranges.

At least, our data contained beta peaks which is significantly different from the 1/f pink noise. In the alpha range, although there were detectable peaks in the LFPs, we could not exclude the possibility that those were 1/f pink noise contained in the LFPs. Therefore, as for the “information encoding” perspective, we focused on the beta range (13-30 Hz). To emphasize these features, we explained these:

L.180: To identify task-related LFPs, we calculated peak frequencies of the spectral power density of all baseline-subtracted LFPs in the four regions and found that many peaks were in the beta range (13-30 Hz) (**Supplementary Fig. 6a**). The grand average of the power spectrum of all baseline-subtracted LFPs also showed beta oscillations in the four regions (**Supplementary Fig. 6b**). Based on these findings, we focused on beta oscillations in the FCS network.

On the other hand, as the Reviewer wisely pointed out, there were indeed two peaks in the alpha (5-12 Hz) and beta ranges (13-30 Hz) in the coherence and Granger analyses (**Supplementary Fig. 11**). In the gamma range (35-100 Hz), both coherence and Granger analyses did not produce significant peaks.

To calculate the GCI, we thus focused on alpha and beta ranges and explicitly mentioned this in the main text:

L.294: First, we examined the frequency band in which the interareal coherence was observed. Our analysis of coherence in all pairs ($n = 3172$) showed peaks around 7 Hz and 20-25 Hz, but no peaks in the gamma range (**Supplementary Fig. 11a**). We thus focused on the coherence in the alpha and beta ranges (5-30 Hz).

In the light of the Reviewer’s suggestion, we separate the alpha and beta influence of the GCI (Granger Causal Influences), and directly assessed whether the beta wave component plays a significant role in producing negative bias in the decision-making.

As indicated above, it was observed that effective stimuli not only attenuated the top-down signal from the dlPFC through beta oscillations but also enhanced bottom-up signals during the stimulus period in the alpha range.

L.310: We next tested the prediction that sgACC microstimulation could attenuate top-down influences from the dlPFC. To calculate directional influences in the network, we compared the difference of GCIs across pairs of LFPs (**Fig. 6b, c**). The microstimulation significantly reduced the top-down influence in the beta oscillation while enhancing the bottom-up influence mediated by alpha oscillation in the FCS network.

Specifically, in the alpha range, GCIs from the sgACC and striatum indicated a pronounced strengthening of the bottom-up influences. In the beta range, the top-down influences originating from the dlPFC were significantly attenuated (**Fig. 6b**).

Thank you so much for your comment. We think the manuscript is much improved.

*Comment 2-7: While the authors primarily concentrated on analyzing causality between regions using GCI, which is valuable for understanding temporal relationships, it is also worth exploring the similarity of information processing among these structures. (1) I propose conducting “representational similarity analysis” with the data, as outlined by Kriegeskorte and Bandettini (2008, *Frontiers in Systems Neuroscience*). This analysis can offer insights into how structures within the FCS network collectively process information in the context of decision-making.*

Reply: We again thank the Reviewer for their comments. Following the suggestions of the Reviewer, we conducted a ‘representational similarity analysis (RSA)’ to statistically compare how dlPFC, sgACC, pACC, and striatum differ in their responses to cue stimuli. Referring to the overview provided by Kriegeskorte and Bandettini (2008, *Frontiers in Systems Neuroscience*) (Kriegeskorte et al., 2008), we utilized the following toolbox (Nili et al. 2014, *PLoS Comput Biol*) (Nili et al., 2014) for the analysis. In the Results section, we added:

L.228: We further employed a representational similarity analysis (RSA)(Kriegeskorte et al., 2008; Nili et al., 2014) to explore the similarity of information processing among the four structures we recorded (**Supplementary Fig. 8a, b**).

This analysis aims to show insights into how the structural elements within the FCS network collectively engage in information processing during decision-making. We acknowledge the significance of comparing the similarities in information processing across various regions of the FCS network. To address this aspect, we conducted a ‘representational similarity analysis (RSA)’ by generating activity vectors for each region in response to cue conditions, organized into 8-by-8 matrices. The RSA methodology aligns with our approach of measuring similarity in the MDS space. Detailed descriptions of these procedures can be found in the Methods section:

L.675: We conducted representational similarity analysis (RSA) to statistically compare how dlPFC, sgACC, pACC, and striatum differ in their responses to cue stimuli by utilizing the Matlab toolbox(Kriegeskorte et al., 2008). This analysis can provide insights into how the structure within the FCS network collaboratively processes information in the context of approach-avoidance decision-making. In

the Ap-Av task, the lengths of the reward and the air-puff bars were continuous. Therefore, to categorize the experimental conditions, we discretized the sizes of the reward and air puff into eight bins, resulting in 64 ($= 8 \times 8$) cue stimuli. To determine the order of experimental conditions, we decoded the utility by regressing decision patterns in the decision-making model (**Supplementary Fig. 8a**), and ranked the experimental conditions based on utility. Magnitudes of beta oscillation were measured while the monkeys were exposed to these 64 experimental conditions. For each brain region of interest (i.e., sgACC, striatum, pACC and dlPFC), the regional activity pattern was estimated by the population activity of beta responses recorded for each experimental condition. A dissimilarity in representation was computed for each pair of activity patterns and put into a representational dissimilarity matrix (RDM) (**Supplementary Fig. 8b**).

L.687: The dissimilarities between the activity patterns can be considered distances in the multivariate response space, and the RDM describes the geometry of the representation, serving as a signature that can be compared between different brain regions. To visualize the relationship among the representations of four brain regions, we performed multi-dimensional scaling of the four RDMs (**Supplementary Fig. 8c**) and the correlation analyses (Kendall's tau and Spearman's test) (**Supplementary Fig. 8d**).

In this manner, we successfully uncovered variations in representations across different regions. The outcomes of this analysis have been succinctly presented in the main text as:

L.229: The analysis showed that cortical areas (sgACC, dlPFC, and pACC) exhibited distinct activation patterns for high utility, whereas the striatum showed varied activation for middle and low utilities,

indicating contrasting regional representations between the striatum and cortices (**Supplementary Fig. 8c, d**). The RSA thus evaluated the striatum as different from the three cortical areas, similar to the clustering procedure results that showed the striatum contained a substantial number of the N group (**Fig. 3e, f**). Thus, the RSA repeatedly showed the regional specificities in the distribution of different types of beta responses in the FCS network.

We appreciate the insights provided by the Reviewer.

Comment 2-8:

In Figure 4b, it is evident that the beta representation of the P group decreased. However, I am curious about the behaviour of the other groups, particularly those that are not involved in the P or N categories. (1) Were there any notable changes observed in these additional groups?

Reply: In **Fig. 4c**, we investigated changes in the number of data points classified not only in the P group but also in other groups. Interestingly, we found an increase in the cyan group, showing a negative correlation with airpuff offer, during the *Follow-up* period. Such changes were not observed during non-effective sessions. We added the following explanation in the Results section:

L.265: The cyan group increased the proportion in the effective *Follow-up* block ($P < 0.05$). No significant change was observed in other groups ($P > 0.05$). These results suggest that effective microstimulation of the sgACC could induce a negative bias in decision-making by changing the representation of the beta oscillation in the FCS network, particularly for the P group.

We are grateful for the Reviewer's keen insight, which allowed us to discover these new dynamics.

Comment 2-9:

In Figures 5 and 6, the focus is on highlighting unidirectional connections. However, it's well-established that loop circuits exist within cortico-striatal and cortico-cortical networks, potentially generating backward information flow. For instance, Figure 5a suggests information flow from the striatum to sgACC. (1) Have the authors conducted any analyses to assess the significance of these backward information flows?

Reply:

We appreciate your keen insight. We have newly computed the Granger Causality Index (GCI) during the stimulation (*Stim-on* block).

L.310: We next tested the prediction that sgACC microstimulation could attenuate top-down influences from the dIPFC. To calculate directional influences in the network, we compared the difference of GCIs across pairs of LFPs (**Fig. 6b, c**). The microstimulation significantly reduced the top-down influence in the beta oscillation while enhancing the bottom-up influence mediated by alpha oscillation in the FCS network. Specifically, in the alpha range, GCIs from the sgACC and striatum indicated a pronounced strengthening of the bottom-up influences. In the beta range, the top-down influences originating from the dIPFC were significantly attenuated (**Fig. 6b**).

In **Fig. 6b**, it has become evident that, in addition to the attenuation of top-down alpha waves during stimulation, there is a significant emergence of reverse alpha waves. These bottom-up directional information flows were observed during stimulation.

Comment 2-10: 2. Minor comments

a. In the Introduction, the term ‘cognitive region’ is employed, but its definition appears to be more functionally oriented rather than anatomically precise. To enhance clarity, it would be beneficial for the authors to provide a more detailed explanation of the functions associated with the dorsolateral prefrontal cortex (dIPFC). This should encompass its roles in working memory, emotion regulation, and other pertinent cognitive functions to establish it as a cognitive region.

Reply: We thank the Reviewer for bringing this to our attention. As suggested, we have provided an explanation in the Introduction regarding the functions associated with the dorsolateral prefrontal cortex (dIPFC), incorporating cognitive functions such as switching attention, working memory, and maintaining abstract rules, as well as its role in emotion regulation.

L.35: The dorsolateral prefrontal cortex (dlPFC), which has long been implicated in cognitive functions, such as switching attention (Buschman & Miller, 2007; Womelsdorf et al., 2014), working memory (Lundqvist et al., 2016; Salazar et al., 2012), and categorical learning (Antzoulatos & Miller, 2014), is also thought to be a center of emotion regulation (Etkin et al., 2015; Ochsner & Gross, 2005). The interareal interaction between the cognitive dlPFC and the limbic system is a possible critical factor for the protective mechanism against the development of MDD (Cash et al., 2021; Joormann & Gotlib, 2010).

We appreciate your valuable comments.

Comment 2-11:

b. The increase in pupil size induced by microstimulation prompts questions about its potential implications, particularly in aspect of the salience network theory. Do the authors have any insights into the significance of this observed pupil response and its alignment with the principles of the salience network theory?

Reply: We are grateful to the Reviewer for raising this crucial point. The salience network, known for its connectivity between the anterior insular and anterior cingulate cortex, plays an important role in detecting salience associated with signal detection (Menon, 2011). In our study, microstimulation of the sgACC resulted in an observed increase in skin conductance arousal levels. Similar physiological response changes linked to sgACC activation have been documented in other studies (Alexander et al., 2020), suggesting an associated rise in arousal level (Bradley et al., 2008). Given the established role of the salience network in regulating such physiological indicators, it is plausible that sgACC activity correlates with saliency. These correspondences have been explicitly outlined in the Results section:

L.141: We observed a significant increase in pupil size without inducing any eye movements during the microstimulation (**Supplementary Fig. 3b-e**). Consistent with previous studies (Alexander et al., 2020), these findings suggest that the activity of the sgACC plays a causal role in regulating physiological responses. This supports the idea that the sgACC may be a key node in regulating saliency and arousal (Bradley et al., 2008), essential for recognizing and responding to important stimuli (Menon, 2011).

We appreciate your valuable comments.

Comment 2-12:

c. It is necessary to engage in a discussion of the study's results within the broader context of the functions associated with beta waves, which have been extensively studied across various neuroscience fields. These functions encompass motor control, attention, decision-making, anxiety modulation, and facilitating communication between different brain regions. Such a discussion not only aids in understanding the

potential role of the fronto-cingulo-striatal network in decision-making but also sheds light on the implications of microstimulation within this context. Integrating the findings into this comprehensive framework would provide valuable insights and enrich the interpretation of the study's results.

Reply: As you suggested, functions related to beta oscillations are widely known to be related to motor control and attention. Furthermore, there are indications that beta waves are implicated in decision-making, necessitating a summary of these previous studies. In accordance with your suggestion, we have added the following sentence in the Introduction:

L.59: The function of beta oscillations has been extensively explored, uncovering their involvement in motor control(Bauer et al., 2006; Pfurtscheller & Lopes da Silva, 1999), attention(Bauer et al., 2006), and decision-making(Haegens et al., 2011).

In the Discussion, we also mentioned that beta oscillation plays an important role in multiple functions as:

L.368: Beta oscillations have traditionally been associated with motor control(Pfurtscheller & Lopes da Silva, 1999) and attention(Bauer et al., 2006; Buschman & Miller, 2007; Siegel et al., 2008). More recently, beta oscillations have also been implicated in cognitive functions, including working memory(Salazar et al., 2012), somatosensory decision-making(Haegens et al., 2011), and negative bias in value judgment(Amemori et al., 2018). Diminishing beta responses encoding positive utility appears comparable to the blunted anticipation to reward induced by over-activation of the sgACC reported in previous studies(Alexander et al., 2019; Alexander et al., 2020; Wallis et al., 2017; Wallis et al., 2019). Our results indicated that the beta-band oscillatory activity, which could serve as the mediator of information flow in the FCS network, encoded positive utility and was particularly reduced by the experimentally-induced negative decision-making.

Thank you so much for pointing this out.

Comment 2-13:

d. P groups were found in most structures, whereas N groups were predominantly observed in the striatum. Do you have any insights into the potential causes of this pattern and its significance?

Reply: We thank you to encourage us to consider this point. We performed representational similarity analysis (RSA in **Supplementary Fig. 8**) and decoding of beta representation (**Supplementary Fig. 9**). The RSA showed that, while the cortical areas (sgACC, dIPFC, pACC) showed differential activation for high utility, while the striatum also showed differential activation for middle and low utilities. These results

showed a contract feature of the regional representation between the striatum and cortices.

L.228: We further employed a representational similarity analysis (RSA)(Kriegeskorte et al., 2008; Nili et al., 2014) to explore the similarity of information processing among the four structures we recorded (**Supplementary Fig. 8a, b**). The analysis showed that cortical areas (sgACC, dlPFC, and pACC) exhibited distinct activation patterns for high utility, whereas the striatum showed varied activation for middle and low utilities, indicating contrasting regional representations between the striatum and cortices (**Supplementary Fig. 8c, d**). The RSA thus evaluated the striatum as different from the three cortical areas, similar to the clustering procedure results that showed the striatum contained a substantial number of the N group (**Fig. 3e, f**). Thus, the RSA repeatedly showed the regional specificities in the distribution of different types of beta responses in the FCS network.

Similarly, decoding of the beta representation further showed that the cortical regions mainly encoded P and other groups, while the striatum beta also contains channels characterized by N groups. We summarized the features in the Results as:

L.236: Additionally, we performed regression analyses to determine the representation of the beta responses (**Supplementary Fig. 9**).

and summarized as:

L.245: These results showed that the P groups exhibited activation for positive expectation of reward and utility, while the N group encoded negative expectation of reward and utility. Taken together, these analyses suggest that the P group might have a relatively prominent role in communicating the Ap-Av decision variable across different brain regions of the FCS network.

As for the “*any insights into the potential causes of this pattern and its significance,*” we mentioned it in Discussion as:

L.375: Further, in our previous study(Amemori et al., 2018), striatal microstimulation was observed to have no effect on beta responses encoding positive utility but instead heightened beta responses associated with negative utility. This indicated that a subset of striatal beta oscillations specifically represented negative utility, and activating the striatum could enhance the processing of negative value. Conversely, the present findings suggest that circuits influenced by sgACC activity represent beta oscillations responsive to positive utility. The activation of sgACC, in turn, appears to induce the suppression of positive values encoded by these circuits.

Thank you so much for your suggestion.

Comment 2-14:

e. Have you observed the suppression of beta wave prior to the movement?

Reply: Yes. Monkeys reported their decisions by moving the joystick during the response period (after the decision period). Beta suppression induced by movement was temporarily separated from the decision-related beta modulation, as shown in the time course of population activities shown in **Fig. 3**.

Comment 2-15:

f. I would appreciate further clarification regarding the term ‘non-effective sessions.’ The use of a 5% discrimination threshold is a valuable criterion for defining the effectiveness of microstimulation. Does this imply that microstimulation led to decision changes in certain sessions while not eliciting such changes in others, even when applied to the same brain region?

Reply: Yes. We further found a positive correlation of the feature of recorded neurons at the stimulation sites (**Fig. 2** and **Supplementary Fig. 4**). The size of bins that showed a significant correlation was 1 mm (-0.5 mm to +0.5 mm) in the depth of the electrode. These results suggest that the effects induced by the microstimulation were not determined by the broad region (i.e., sgACC area) but by the function of the narrower areas stimulated at each experiment. We explicitly mentioned these correlations and concluded as follows.

L.160: These results suggest that activation of ‘avoidance neurons’ and ‘airpuff (+) neurons’ in the sgACC could serve as part of the network that is causally involved in negative bias in conflict decision-making.

We appreciate your feedback.

Comment 2-16:

g. Have the authors formulated any hypotheses or insights regarding the mechanisms through which microstimulation generates a negative bias in decision-making? Is it believed that microstimulation activates or deactivates the sgPFC? Additionally, what are the expectations or predictions if the sgACC were to be inactivated in the context of the study?

Reply: We hypothesized that the microstimulation induces activation in localized neural circuits of sgACC, leading to changes in decision-making. The basis for this hypothesis lies in the observation that the recorded neural activity at effective stimulation sites shows a positive correlation with the stimulation effect (**Fig. 2** and **Supplementary Fig. 4**). Furthermore, this hypothesis is also based on the correlation between sgACC activation and the pathophysiology of MDD, as explicitly stated in the Introduction.

L.72: In MDD patients, the sgACC exhibited elevated metabolic activity, which decreased with successful antidepressant treatment(Mayberg et al., 1999).

Furthermore, these results align consistently with various previous marmoset experimental findings, as explicitly outlined in the discussion section, for instance.

L.370: Diminishing beta responses encoding positive utility appears comparable to the blunted anticipation to reward induced by over-activation of the sgACC reported in previous studies(Alexander et al., 2019; Alexander et al., 2020; Wallis et al., 2017; Wallis et al., 2019).

In human clinical studies as well, the activation of dIPFC through TMS, alleviating the depressive symptoms, suggests a decrease in sgACC activity, as explicitly stated in the Introduction.

L.47: Specifically, the interaction among FCS network has been explored through transcranial magnetic stimulation (TMS) on the dIPFC, commonly used for MDD treatment(Cash et al., 2021). The antidepressant effectiveness of the dIPFC activation is linked to their anticorrelated activities of the subgenual anterior cingulate cortex (sgACC)(Fox et al., 2012).

We mentioned that we targeted sgACC for microstimulation, because of the reasons above. We stated in Introduction as:

L.71: The sgACC was selected as the target for microstimulation due to its critical role in emotional modulation in MDD.

To address the question of “*what are the expectations or predictions if the sgACC were to be inactivated,*” we formulated a hypothesis suggesting that the activation of the dlPFC regulates (or suppresses) sgACC activity to restore a normalized state resembling MDD. This possibility was explored and discussed as:

L.410: Concerning the mechanism of how sgACC microstimulation disrupts top-down control, the interplay between sgACC and dlPFC is considered critical. Previous clinical research on patients with MDD consistently reports a negative correlation in activities between cognitive and limbic regions (Cash et al., 2021). Exploring interactions within the FCS network, the TMS on the dlPFC has been recognized for its antidepressant effects and is consistently associated with anticorrelated activities with sgACC (Fox et al., 2012). These findings, aligning with a marmoset study (Alexander et al., 2020), demonstrate that sgACC activation disrupts connectivity between sgACC and dlPFC. Our study further illustrates that the interplay between cognitive and limbic regions may be mediated by alpha and beta-range oscillations, with sgACC activation disturbing signaling within the FCS network.

Thank you for your feedback.

Comment 2-17:

h. Page 6, line 117: It is mentioned that 25 out of 38 sites were non-effective, the term ‘overall’ may appear too strong.

Reply: We agree the Reviewer’s comment and changed “overall” to “the part of”.

Comment 2-18:

i. Page 5, line 101: There is a typo in the word “Recording”.

j. Page 9, line 192: There is a typo in the word “tinthe”.

k. Page 11, line 234: Should it be ‘Stim-on’ instead of ‘Stim-off’?

Reply: You are right! We corrected as you suggested. Thank you so much for the corrections.

Comment 2-21:

l. Page 12, line 270: It appears that ‘microstimulation’ should be substituted for ‘a negative bias in decision-making’.

Reply: We revised as you suggested.

Comment 2-22: m. Page 14, line 306: Bringing up the visuomotor system in this sentence appears to be

unrelated or out of context.

Reply: In the relationship between cognitive and visual system, beta oscillation has been implicated in hierarchical information processing. We changed the word from “visuomotor” to “the interaction between cognitive and visual system.” Thank you for your suggestion.

Comment 2-23: n. Page 22, line 464: There is a typo in the word “significant”.

Reply: We corrected as you suggested. Thank you so much for the corrections.

Reviewer #3 (Remarks to the Author):

The authors report a study in which they carried out microstimulation and neurophysiology recordings while monkeys engaged in an approach avoid task. In the task, the monkeys accepted or rejected offers that combined air puff and juice rewards. Microstimulation was done in the sub-genual cingulate. Most sites induced an increase in avoidance behavior. The authors also examined evoked LFP responses in the beta band and found that an important dimension driving beta LFP responses was the approach/avoid decision. Interestingly, microstimulation decreased the number of sites at which there was a larger response in approach trials. Microstimulation also decreased the functional connectivity in the network. This study addresses an important question about how microstimulation affects both behavior and neural activity across frontal-striatal networks. The approach avoid task is one of the few tasks that allows examination of the network underlying important decisions that may be related to depression or anxiety. Overall, this study makes an important contribution to understanding both the neural systems underlying this important decision making process, and how it may be affected by microstimulation. I do have a number of questions, however, which should be addressed.

Reply: We appreciate the positive comments of the Reviewer indicating that the study makes an important contribution to understanding both the neural systems underlying this important decision-making process, and how it may be affected by microstimulation. We hope that we have answered all concerns as detailed below.

Comment 3-1:

It is difficult to determine the recording and stimulation locations from Figure 1e. A figure showing the locations on coronal sections of the brain would be useful. For example, it is not clear what the pACC site is? Also, it is important to show clearly which region of the striatum is being recorded from, given the very different connectivity of the dorsal and ventral striatum.

Reply: We thank the Reviewer for pointing out this important aspect. We added another figure to clearly showed the recording locations in **Fig. 1f**.

Comment 3-2:

2. In the Introduction, the circuit organization is described from a cognitive-limbic perspective, and the striatum is considered limbic. However, there are both cognitive and limbic regions in the striatum. dlPFC projects dorsally, and sgAcc projections somewhat ventrally into the striatum. Thus, the circuit organization as described in the intro is confusing. This should be clarified, and clarified in relation to the actual recording locations, as described in comment 1.

Reply: We express our gratitude to the Reviewer for highlighting these issues. Regarding the inclusion of both cognitive and limbic territories in the striatum, we acknowledge the Reviewer's comment. In the revised manuscript, we have taken great care to distinguish between 'cingulo-striatal regions,' identified as limbic, and 'fronto-striatal regions,' characterized as cognitive, in the FCS network. This clarification has been incorporated into the Introduction section. Specifically, for the limbic territory of the striatum, we provided the following explanation:

L.39: Especially, the fronto-cingulo-striatal (FCS) network, which includes the fronto-striatal circuit as the cognitive system(Chrysikou et al., 2022) and the cingulo-striatal circuit as the limbic system(Gabbay et al., 2013; Pizzagalli et al., 2001), has been implicated in so-called negative processing bias(Grimm et al., 2008; Pizzagalli & Roberts, 2022) in MDD patients(Mayberg, 2009; Xu et al., 2021), who tend to react negatively to emotionally evocative stimuli(Scheele et al., 2013).

For cognitive territory of striatum, we explained as:

L.42: Fronto-cingulate(Drevets & Raichle, 1998) and fronto-striatal(Furman et al., 2011; Heller et al., 2013) interactions have been implicated in both cognition and emotion, highlighting the potential role of dlPFC in emotional regulation.

To address the question of '*what exactly is included in the fronto-cingulate-striatal network*,' we based our selection on the clinical hypothesis of MDD-related circuitry. As for the **clarification in relation to the actual recording locations**, we clarified the anatomical connections between pACC and striatum through previous anatomical studies(Amemori et al., 2020), as well as the established fronto-striatal connectivity reported in earlier articles(Ferry et al., 2000). Moreover, reciprocal connections between pACC and sgACC were confirmed in previous research articles(Amemori et al., 2021).

L.82: Our targets within the FCS network included the dlPFC as the cognitive area and the pACC and sgACC as cingulate areas known to have reciprocal connectivity(Amemori et al., 2021). Additionally, we targeted the dorsal part of the striatum(Amemori et al., 2018), which is recognized for receiving projections

from the dlPFC(Ferry et al., 2000) and pACC(Amemori et al., 2020).

Thanks to the Reviewer's comment, we thus have clarified that they are closely interconnected based on the forward tracing of the FCS network.

Comment 3-3:

3. When within the trial was stimulation applied? This should be mentioned in the results, since this is a results first, methods last journal.

Reply: We thank the Reviewer for pointing out these problems and explicitly mentioned it in Result as:

L.125: We performed microstimulation for 1 s from the start of the decision period at every trial in the *Stim-on* block.

Thank you for your feedback.

Comment 3-4:

4. Increased Av could follow from more consistently choosing avoid for the avoid regions, or shifting the boundary. Which did the stimulation affect?

Reply: We appreciate the Reviewer for bringing these concerns to our attention. In response to this query, we investigated the cumulative impact of microstimulation throughout the *Stim-on* block. Our approach involved a comprehensive analysis of the temporal dynamics of stimulation effects. Notably, we observed that behavioral changes induced by the stimulation showed a temporary accumulation of effects, with their significance reaching a peak in the latter stages of the stimulation block.

As illustrated in **Supplementary Fig. 13a**, microstimulation progressively shifted the decision boundary, leading to a gradual increase in avoidance choices. Similar characteristics were reported in our previous studies involving pACC(Amemori & Graybiel, 2012) and striatal microstimulation(Amemori et al., 2018),

and these features were consistently robust in these microstimulation experiments. We explained these features in the main text:

L.317: We further explored the causal relationship between the network-level changes and the behavioral alterations. Notably, the behavioral changes induced by the stimulation exhibited temporal accumulation (**Supplementary Fig. 13a**), a feature consistently reported in our previous studies (Amemori et al., 2018; Amemori & Graybiel, 2012).

Thank you for your feedback.

Comment 3-5:

5. Fisher's exact test was used for Figure 1G, but this was not used for the session-level analyses, which instead used a change of 5%. How many sessions showed individually significant differences using Fisher's exact test?

Reply: We appreciate the Reviewer's insightful comments. In our analysis using Fisher's exact test, we found that 19 out of 43 sessions (44.2%) exhibited statistical significance, indicating that the size of $\Delta V - \Delta A_p$ was larger than zero. Notably, even in the 'control' sessions **without microstimulation**, subtle changes were observed in 11 out of 71 sessions (15.5%) as depicted in **Supplementary Fig. 2b**.

To determine a reliable threshold for characterizing the stimulation effect, we considered that a simple comparison between *Stim-off* and *Stim-on* blocks may be inadequate. To ensure a robust definition and minimize the potential for behavioral changes unrelated to stimulation, we conducted a comparative analysis between stimulation and control sessions (**Supplementary Fig. 2b**). The 5% threshold was subsequently established, ensuring a false positive rate of less than 2.8%.

Comment 3-6:

6. In supplemental Fig 6d and e on the MDS analysis, what do i and j refer to? In other words, what is the distance calculated between? Trial conditions?

Reply: We appreciate the Reviewer for highlighting these concerns. As detailed in **Fig. 3a-b**, we computed a 'beta response' from each recording channel. An illustrative example of a beta response is presented in **Fig. 3c** as:

In **Supplementary Fig. 6**, each i or j represented the beta response recorded from individual recording channel. We added the following explanation:

L.108 of supplementary Information: The color of each element shows the correlation distance ($d_{ij} = 1 - r_{ij}$), where r_{ij} is the cross-correlation between beta response i and response j recorded from each channel.

To define the similarity of beta responses r_{ij} , we calculated the cross-correlation of the two beta responses.

Comment 3-7:

7. The paragraph that begins with, “To examine the autonomic responses induced by effective microstimulation, we conducted an additional experiment in which we applied microstimulation to three negative effective sites while the monkeys performed...” should perhaps be further clarified. The final sentence of the paragraph is clear, but perhaps it would be better to suggest, in the first sentence, not that you are examining autonomic responses, but you want to examine the extent to which the microstimulation drives an aversive affect state directly. It seems to have some affect, so perhaps the point should be that the effect is minor relative to the air puff?

Reply: We appreciate your feedback on this matter, and your observation is accurate. To provide additional clarity, we explicitly state that the effect of microstimulation on the physiological response is significantly smaller compared to the physiological response induced by the mild aversive stimulus of the airpuff. This clarification has been added:

L.147: Because the effect of the microstimulation on the physiological response is significantly smaller than that induced by the airpuff, we concluded that the reflexive aversive reaction to microstimulation was too weak to influence Ap-Av decision-making.

Thank you for your feedback.

Comment 3-8:

8. In the sentence, “We defined an LFP as task-related if it showed a significant difference in beta power

($P < 0.05$, z test, Bonferroni corrected)” what is being Bonferroni corrected? Frequency bands? How many were tested?

Reply: We thank the Reviewer for highlighting these issues. In response to your concerns, we have addressed them by correcting for the number of frequency bands. We utilized 1-Hz bins and applied Bonferroni correction with $n = 25$, given the repetition of statistical analysis in the 5-30 Hz range. Additional sentences have been incorporated into the **Method** section for clarification:

L.654: To examine the difference between population spectrum of two conditions, we employed 1-Hz bins and performed t-test between two populations. We addressed the concerns of multiple comparison by correcting for the number of frequency bands. We applied Bonferroni correction with $n = 25$, as we repeated the statistical analysis in the 5-30 Hz range.

Thank you for your insightful comments.

Comment 3-9:

9. In the sentence, “Among the eight groups, the black and gray groups” I assume gray should say green or dark green?

Reply: We thank the Reviewer for pointing out these problems. We changed the color to gray.

Comment 3-10:

10. Again it’s not clear which part of the striatum was being recorded from, but it’s interesting that the striatum did not perhaps have a strong representation of the P group in the analysis in Figure 3.

Reply: We thank the Reviewer for pointing out these problems. As we addressed in *Comments 3-6*, we explained as:

L.83: Additionally, we targeted the dorsal part of the striatum (Amemori et al., 2018), which is recognized for receiving projections from the dIPFC (Ferry et al., 2000) and pACC (Amemori et al., 2020).

Thank you for pointing out this.

Comment 3-11:

11. Page 11, line 234, Stim-off should probably read stim-on.

Reply: We thank the Reviewer for pointing out these problems. We corrected as you suggested.

Comment 3-12:

12. It would be useful to have a more detailed discussion of how the directed coherence is calculated by the toolbox, since coherence is not direction.

Reply: We thank the Reviewer for pointing out these problems. As you pointed out, coherence does not have any directionality. We deleted the word “Directed coherence,” and instead, we calculated GC.

Comment 3-13:

13. In Fig. 6, an absence of significant interactions following stimulation does not show that the interactions were significantly difference with and without stimulation. Statistics should be done to directly test these.

Reply: We appreciate the Reviewer for highlighting these concerns. We took this matter seriously and revised our approach. Initially, our intention was to compare two distinct *Stim-off* periods before and after effective stimulation to minimize potential stimulation artifact influence during the *Stim-on* period. In the revised manuscript, we conducted a meticulous examination of the LFPs, ensuring the successful removal of stimulation artifacts using the outlined method in the **Methods** section:

L.629: To remove electrical stimulation artifacts from the raw 32 kHz-sampled files, we used linear interpolation between the time points 50 ms and 1.5 ms after the onset of the stimulation trigger pulse. However, when the recording electrode was very close to the stimulating electrode, we found that the amplifier sometimes took longer to settle into a usable range than the interval between stimulation pulses, which resulted in a distorted signal throughout the stimulation period. Therefore, we did not attempt to remove stimulation artifacts from these channels.

After eliminating the artifact caused by the stimulation, we conducted GCI directionality analyses (**Supplementary Fig. 12b**) to derive DAI (directional asymmetry index) (**Fig. 6b**).

Remarkably, microstimulation significantly reduced the ‘top-down’ influence in the beta frequency range while concurrently enhancing the ‘bottom-up’ influence mediated by alpha waves. From the GCI during stimulation, we derived the directional orientation of the signals, illustrated in **Figure 6b**, with diagrams depicting the DAI. During stimulation, there was a significant attenuation of top-down signals through beta waves, and concurrently, alpha waves exhibited a strengthening of bottom-up signals. This comparison directly confirmed the reduction in beta waves due to stimulation. We explained these results as:

L.310: We next tested the prediction that sgACC microstimulation could attenuate top-down influences from the dIPFC. To calculate directional influences in the network, we compared the difference of GCIs across pairs of LFPs (**Fig. 6b, c**). The microstimulation significantly reduced the top-down influence in the beta oscillation while enhancing the bottom-up influence mediated by alpha oscillation in the FCS network. Specifically, in the alpha range, GCIs from the sgACC and striatum indicated a pronounced strengthening of the bottom-up influences. In the beta range, the top-down influences originating from the dIPFC were significantly attenuated (**Fig. 6b**).

We appreciate the Reviewer’s insightful comment which drastically improves the quality of the manuscript.

References:

- Aizenstein, H. J., Butters, M. A., Wu, M., Mazurkewicz, L. M., Stenger, V. A., Gianaros, P. J., Becker, J. T., Reynolds, C. F., 3rd, & Carter, C. S. (2009). Altered functioning of the executive control circuit in late-life depression: episodic and persistent phenomena. *Am J Geriatr Psychiatry, 17*(1), 30-42. <https://doi.org/10.1097/JGP.0b013e31817b60af>
- Alexander, L., Gaskin, P. L. R., Sawiak, S. J., Fryer, T. D., Hong, Y. T., Cockcroft, G. J., Clarke, H. F., & Roberts, A. C. (2019). Fractionating blunted reward processing characteristic of anhedonia by over-activating primate subgenual anterior cingulate cortex. *Neuron, 101*(2), 307-320. <https://doi.org/10.1016/J.NEURON.2018.11.021>
- Alexander, L., Wood, C. M., Gaskin, P. L. R., Sawiak, S. J., Fryer, T. D., Hong, Y. T., McIver, L., Clarke, H. F., & Roberts, A. C. (2020). Over-activation of primate subgenual cingulate cortex enhances the cardiovascular, behavioral and neural responses to threat. *Nature Communications, 11*(1), 5386-5386. <https://doi.org/10.1038/s41467-020-19167-0>
- Amemori, K., Amemori, S., Gibson, D. J., & Graybiel, A. M. (2018). Striatal microstimulation induces persistent and repetitive negative decision-making predicted by striatal beta-band oscillation. *Neuron, 99*(4), 829-841. <https://doi.org/10.1016/j.neuron.2018.07.022>
- Amemori, K., Amemori, S., & Graybiel, A. M. (2015). Motivation and affective judgments differentially recruit neurons in the primate dorsolateral prefrontal and anterior cingulate cortex. *Journal of Neuroscience, 35*(5), 1939-1953. <https://doi.org/10.1523/JNEUROSCI.1731-14.2015>
- Amemori, K., & Graybiel, A. M. (2012). Localized microstimulation of primate pregenual cingulate cortex induces negative decision-making. *Nature Neuroscience, 15*(5), 776-785. <https://doi.org/10.1038/nn.3088>
- Amemori, S., Amemori, K., Yoshida, T., Papageorgiou, G. K., Xu, R., Shimazu, H., Desimone, R., & Graybiel, A. M. (2020). Microstimulation of primate neocortex targeting striosomes induces negative decision-making. *European Journal of Neuroscience, 51*(3), 731-741. <https://doi.org/10.1111/ejn.14555>
- Amemori, S., Graybiel, A., & Amemori, K. (2021). Causal evidence for induction of pessimistic decision-making in primates by the network of frontal cortex and striosomes [Hypothesis and Theory]. *Frontiers in Neuroscience, 15*, 649167. <https://doi.org/10.3389/fnins.2021.649167>
- Antzoulatos, E. G., & Miller, E. K. (2014). Increases in functional connectivity between prefrontal cortex and striatum during category learning. *Neuron, 83*(1), 216-225. <https://doi.org/10.1016/J.NEURON.2014.05.005>
- Bauer, M., Oostenveld, R., Peeters, M., & Fries, P. (2006). Tactile spatial attention enhances gamma-band activity in somatosensory cortex and reduces low-frequency activity in parieto-occipital areas. *Journal of Neuroscience, 26*(2), 490-501. <https://doi.org/10.1523/JNEUROSCI.5228-04.2006>
- Bradley, M. M., Miccoli, L., Escrig, M. A., & Lang, P. J. (2008). The pupil as a measure of emotional arousal and autonomic activation. *Psychophysiology, 45*(4), 602-607. <https://doi.org/10.1111/j.1469-8986.2008.00654.x>
- Buschman, T. J., & Miller, E. K. (2007). Top-down versus bottom-up control of attention in the prefrontal and posterior

- parietal cortices. *Science*, 315(5820), 1860-1862. <https://doi.org/10.1126/science.1138071>
- Cash, R. F. H., Weigand, A., Zalesky, A., Siddiqi, S. H., Downar, J., Fitzgerald, P. B., & Fox, M. D. (2021). Using brain imaging to improve spatial targeting of transcranial magnetic stimulation for depression. *Biological Psychiatry*, 90(10), 689-700. <https://doi.org/10.1016/j.biopsych.2020.05.033>
- Chryssikou, E. G., Wing, E. K., & van Dam, W. O. (2022). Transcranial direct current stimulation over the prefrontal cortex in depression modulates cortical excitability in emotion regulation regions as measured by concurrent functional magnetic resonance imaging: An exploratory study. *Biological Psychiatry: Cognitive Neuroscience and Neuroimaging*, 7(1), 85-94. <https://doi.org/10.1016/J.BPSC.2019.12.004>
- Clarke, H. F., Horst, N. K., Roberts, A. C., Clarke, H. F., Horst, N. K., Roberts, A. C., Hf Clarke, N. K. H. A. C. R., Clarke, H. F., Horst, N. K., & Roberts, A. C. (2015). Regional inactivations of primate ventral prefrontal cortex reveal two distinct mechanisms underlying negative bias in decision making. *Proceedings of the National Academy of Sciences of the United States of America*, 112(13), 4176-4181. <https://doi.org/10.1073/pnas.1422440112>
- Drevets, W. C., & Raichle, M. E. (1998). Suppression of Regional Cerebral Blood during Emotional versus Higher Cognitive Implications for Interactions between Emotion and Cognition. *Cognition and Emotion*, 12(3), 353-385. <https://doi.org/10.1080/026999398379646>
- Etkin, A., Büchel, C., & Gross, J. J. (2015). The neural bases of emotion regulation. *Nature Reviews Neuroscience*, 16(11), 693-700. <https://doi.org/10.1038/nrn4044>
- Etkin, A., Egner, T., Peraza, D. M., Kandel, E. R., & Hirsch, J. (2006). Resolving emotional conflict: a role for the rostral anterior cingulate cortex in modulating activity in the amygdala. *Neuron*, 51(6), 871-882. <https://doi.org/10.1016/j.neuron.2006.07.029>
- Ferry, A. T., Öngür, D., An, X., & Price, J. L. (2000). Prefrontal cortical projections to the striatum in macaque monkeys: Evidence for an organization related to prefrontal networks. *Journal of Comparative Neurology*, 425(3), 447-470. [https://doi.org/https://doi.org/10.1002/1096-9861\(20000925\)425:3<447::AID-CNE9>3.0.CO;2-V](https://doi.org/https://doi.org/10.1002/1096-9861(20000925)425:3<447::AID-CNE9>3.0.CO;2-V)
- Fox, M. D., Buckner, R. L., White, M. P., Greicius, M. D., & Pascual-Leone, A. (2012). Efficacy of transcranial magnetic stimulation targets for depression is related to intrinsic functional connectivity with the subgenual cingulate. *Biol Psychiatry*, 72(7), 595-603. <https://doi.org/10.1016/j.biopsych.2012.04.028>
- Furman, D. J., Hamilton, J. P., & Gotlib, I. H. (2011). Frontostriatal functional connectivity in major depressive disorder. *Biol Mood Anxiety Disord*, 1(1), 11. <https://doi.org/10.1186/2045-5380-1-11>
- Gabbay, V., Ely, B. A., Li, Q., Bangaru, S. D., Panzer, A. M., Alonso, C. M., Castellanos, F. X., & Milham, M. P. (2013). Striatum-based circuitry of adolescent depression and anhedonia. *Journal of the American Academy of Child and Adolescent Psychiatry*, 52(6), 628-628. <https://doi.org/10.1016/J.JAAC.2013.04.003>
- Grimm, S., Beck, J., Schuepbach, D., Hell, D., Boesiger, P., Bermppohl, F., Niehaus, L., Boeker, H., & Northoff, G. (2008). Imbalance between left and right dorsolateral prefrontal cortex in major depression is linked to negative emotional judgment: an fMRI study in severe major depressive disorder. *Biological Psychiatry*, 63(4), 369-376. <https://doi.org/10.1016/J.BIOPSYCH.2007.05.033>
- Haegens, S., Nacher, V., Hernández, A., Luna, R., Jensen, O., & Romo, R. (2011). Beta oscillations in the monkey

- sensorimotor network reflect somatosensory decision making. *Proceedings of the National Academy of Sciences of the United States of America*, 108(26), 10708-10713. <http://www.pnas.org/content/108/26/10708.abstract>
- Heller, A. S., Johnstone, T., Light, S. N., Peterson, M. J., Kolden, G. G., Kalin, N. H., & Davidson, R. J. (2013). Relationships between changes in sustained fronto-striatal connectivity and positive affect in major depression resulting from antidepressant treatment. *Am J Psychiatry*, 170(2), 197-206. <https://doi.org/10.1176/appi.ajp.2012.12010014>
- Holmes, A. J., & Pizzagalli, D. A. (2008). Response conflict and frontocingulate dysfunction in unmedicated participants with major depression. *Neuropsychologia*, 46(12), 2904-2913. <https://doi.org/10.1016/j.neuropsychologia.2008.05.028>
- Ironside, M., Amemori, K., McGrath, C. L., Pedersen, M. L., Kang, M. S., Amemori, S., Frank, M. J., Graybiel, A. M., & Pizzagalli, D. A. (2020). Approach-Avoidance conflict in major depressive disorder: Congruent neural findings in humans and nonhuman primates. *Biological Psychiatry*, 87(5), 399-408. <https://doi.org/10.1016/j.biopsych.2019.08.022>
- Joormann, J., & Gotlib, I. H. (2010). Emotion regulation in depression: Relation to cognitive inhibition. *Cognition & Emotion*, 24(2), 281-281. <https://doi.org/10.1080/02699930903407948>
- Kriegeskorte, N., Mur, M., & Bandettini, P. (2008). Representational similarity analysis - connecting the branches of systems neuroscience. *Front Syst Neurosci*, 2, 4. <https://doi.org/10.3389/neuro.06.004.2008>
- Lundqvist, M., Rose, J., Herman, P., Brincat, S. L., Buschman, T. J., & Miller, E. K. (2016). Gamma and Beta Bursts Underlie Working Memory. *Neuron*, 90(1), 152-164. <https://doi.org/10.1016/j.neuron.2016.02.028>
- Margulies, D. S., Kelly, A. M., Uddin, L. Q., Biswal, B. B., Castellanos, F. X., & Milham, M. P. (2007). Mapping the functional connectivity of anterior cingulate cortex. *NeuroImage*, 37(2), 579-588. <https://doi.org/10.1016/j.neuroimage.2007.05.019>
- Mayberg, H. S. (2009). Targeted electrode-based modulation of neural circuits for depression. *Journal of Clinical Investigation*, 119(4), 717-725. <https://doi.org/10.1172/JCI38454>
- Mayberg, H. S., Liotti, M., Brannan, S. K., McGinnis, S., Mahurin, R. K., Jerabek, P. A., Silva, J. A., Tekell, J. L., Martin, C. C., Lancaster, J. L., & Fox, P. T. (1999). Reciprocal limbic-cortical function and negative mood: converging PET findings in depression and normal sadness. *The American journal of psychiatry*, 156(5), 245-253. <https://doi.org/10.1176/AJP.156.5.675>
- Mayberg, H. S., Lozano, A. M., Voon, V., McNeely, H. E., Seminowicz, D., Hamani, C., Schwalb, J. M., & Kennedy, S. H. (2005). Deep brain stimulation for treatment-resistant depression. *Neuron*, 45(5), 651-660. <https://doi.org/10.1016/j.neuron.2005.02.014>
- Menon, V. (2011). Large-scale brain networks and psychopathology: a unifying triple network model. *Trends Cogn Sci*, 15(10), 483-506. <https://doi.org/10.1016/j.tics.2011.08.003>
- Nili, H., Wingfield, C., Walther, A., Su, L., Marslen-Wilson, W., & Kriegeskorte, N. (2014). A toolbox for representational similarity analysis. *PLoS Comput Biol*, 10(4), e1003553. <https://doi.org/10.1371/journal.pcbi.1003553>

- Ochsner, K. N., & Gross, J. J. (2005). The cognitive control of emotion. *Trends in Cognitive Sciences*, 9(5), 242-249. <https://doi.org/10.1016/j.tics.2005.03.010>
- Pfurtscheller, G., & Lopes da Silva, F. H. (1999). Event-related EEG/MEG synchronization and desynchronization: basic principles. *Clinical Neurophysiology*, 110(11), 1842-1857. [https://doi.org/https://doi.org/10.1016/S1388-2457\(99\)00141-8](https://doi.org/https://doi.org/10.1016/S1388-2457(99)00141-8)
- Pizzagalli, D., Pascual-Marqui, R. D., Nitschke, J. B., Oakes, T. R., Larson, C. L., Abercrombie, H. C., Schaefer, S. M., Koger, J. V., Benca, R. M., & Davidson, R. J. (2001). Anterior cingulate activity as a predictor of degree of treatment response in major depression: Evidence from brain electrical tomography analysis. *American Journal of Psychiatry*, 158(3), 405-415. <https://doi.org/10.1176/appi.ajp.158.3.405>
- Pizzagalli, D. A., & Roberts, A. C. (2022). Prefrontal cortex and depression. *Neuropsychopharmacology*, 47(1), 225-246. <https://doi.org/10.1038/s41386-021-01101-7>
- Salazar, R. F., Dotson, N. M., Bressler, S. L., & Gray, C. M. (2012). Content-specific fronto-parietal synchronization during visual working memory. *Science*, 338(6110), 1097-1100. <https://doi.org/10.1126/SCIENCE.1224000>
- Scheele, D., Mihov, Y., Schwederski, O., Maier, W., & Hurlemann, R. (2013). A negative emotional and economic judgment bias in major depression. *European archives of psychiatry and clinical neuroscience*, 263(8), 675-683. <https://doi.org/10.1007/S00406-013-0392-5>
- Siegel, M., Donner, T. H., Oostenveld, R., Fries, P., & Engel, A. K. (2008). Neuronal synchronization along the dorsal visual pathway reflects the focus of spatial attention. *Neuron*, 60(4), 709-719. <https://pubmed.ncbi.nlm.nih.gov/19038226/>
- Wallis, C. U., Cardinal, R. N., Alexander, L., Roberts, A. C., & Clarke, H. F. (2017). Opposing roles of primate areas 25 and 32 and their putative rodent homologs in the regulation of negative emotion. *Proceedings of the National Academy of Sciences*, 114(20), 4075-4084. <https://doi.org/10.1073/pnas.1620115114>
- Wallis, C. U., Cockcroft, G. J., Cardinal, R. N., Roberts, A. C., & Clarke, H. F. (2019). Hippocampal interaction with Area 25, but not Area 32, regulates marmoset Approach-Avoidance behavior. *Cerebral Cortex*, 29(11), 4818-4830. <https://doi.org/10.1093/cercor/bhz015>
- Womelsdorf, T., Ardid, S., Everling, S., & Valiante, T. A. (2014). Burst firing synchronizes prefrontal and anterior cingulate cortex during attentional control. *Current Biology*, 24(22), 2613-2621. <https://doi.org/10.1016/J.CUB.2014.09.046>
- Xu, X., Dai, J., Chen, Y., Liu, C., Xin, F., Zhou, X., Zhou, F., Stamatakis, E. A., Yao, S., Luo, L., Huang, Y., Wang, J., Zou, Z., Vatansever, D., Kendrick, K. M., Zhou, B., & Becker, B. (2021). Intrinsic connectivity of the prefrontal cortex and striato-limbic system respectively differentiate major depressive from generalized anxiety disorder. *Neuropsychopharmacology*, 46(4), 791-798. <https://doi.org/10.1038/S41386-020-00868-5>
- Zikopoulos, B., & Barbas, H. (2012). Pathways for emotions and attention converge on the thalamic reticular nucleus in primates. *Journal of Neuroscience*, 32(15), 5338-5350. <https://doi.org/10.1523/jneurosci.4793-11.2012>

REVIEWERS' COMMENTS

Reviewer #1 (Remarks to the Author):

The reviewers have addressed all my comments very effectively.

Reviewer #2 (Remarks to the Author):

The authors have fulfilled my comments and have much improved their manuscript by adding more analyses and clarification. I am satisfied with this revised version of manuscript.

Reviewer #3 (Remarks to the Author):

The authors have addressed my concerns. I have no further comments.

Point-by-point response to the reviewers' comments

Cingulate microstimulation induces negative decision-making via reduced top-down influence on primate fronto-cingulo-striatal network.

NCOMMS-23-41647B

Reviewer's comments are in blue.

Our replies are in black.

Reviewer #1 (Remarks to the Author):

The reviewers have addressed all my comments very effectively.

Reply: We are pleased that we could address all the concerns.

Reviewer #2 (Remarks to the Author):

The authors have fulfilled my comments and have much improved their manuscript by adding more analyses and clarification. I am satisfied with this revised version of manuscript.

Reply: We are pleased that the Reviewer is satisfied with the revision.

Reviewer #3 (Remarks to the Author):

The authors have addressed my concerns. I have no further comments.

Reply: We are pleased that we could address all the concerns.